# Precision magnetometry exploiting excited state quantum phase transitions

**Qian Wang[1,2] and Ugo Marzolino[3,4,5⋆]**

**1** Department of Physics, Zhejiang Normal University, Jinhua 321004, China
**2** CAMTP-Center for Applied Mathematics and Theoretical Physics,
University of Maribor, Mladinska 3, SI-2000 Maribor, Slovenia
**3** National Institute of Nuclear Physics, Trieste Unit, I-34151 Trieste, Italy
**4** Scuola Normale Superiore, I-56126 Pisa, Italy
**5** University of Trieste, I-34127 Trieste, Italy

⋆ ugo.marzolino@ts.infn.it

## Abstract

Critical behaviour in phase transitions is a resource for enhanced precision metrology. The reason is that the function, known as Fisher information, is superextensive at critical points, and, at the same time, quantifies performances of metrological protocols. Therefore, preparing metrological probes at phase transitions provides enhanced precision in measuring the transition control parameter. We focus on the Lipkin-Meshkov-Glick model that exhibits excited state quantum phase transitions at different magnetic fields. Resting on the model spectral properties, we show broad peaks of the Fisher information, and propose efficient schemes for precision magnetometry. The Lipkin-Meshkov-Glick model was first introduced for superconductivity and for nuclear systems, and recently realised in several condensed matter platforms. The above metrological schemes can be also exploited to measure microscopic properties of systems able to simulate the Lipkin-Meshkov-Glick model.



# 1  Introduction

Estimation theory identifies resources useful for precision measurements of parameters encoded in system states. Of paramount importance is quantum metrology which seeks genuinely quantum enhancements [1–6], and also applies to definitions of measurement units [7,8]: the current standards for electrical resistance and mass rely on the quantum Hall effect and the Planck constant, and nowadays atomic clocks achieve better accuracy than the definition of the second [9]. A versatile metrological framework is magnetometry which benefits from several physical scenarios, from atomic vapor to nuclear magnetic resonance and nitrogen vacancy to semiconductors and superconductors [5]. Industrial applications of magnetometry include non-invasive diagnostics of human organs, non-destructive detection of flaws in materials, localisation of mineral deposits [8].

An experimental challenge consists in reducing systematic errors in order to achieve the so-called shot-noise limit for estimator variances, i.e. $\mathcal{O}(1/N)$, where $N$ is the number of resources, e.g. particles. This scaling can be improved with entangled states that are, however, very fragile with respect to noise [10–12]. It is therefore a great benefit to stabilise such entangled states [13,14], or to achieve quantum enhancements without the burden of preparing entangled probes [6]. It is also desirable to investigate metrological schemes suited for several physical platforms and using different physical phenomena in the search for feasible implementations, so called noisy intermediate-scale quantum (NISQ) technologies [15,16].

A central tool in estimation theory is the quantum Fisher information (QFI). It is the inverse of the best achievable variance of unbiased metrological schemes that employ operations independent from the parameter to be estimated [1–4]. The QFI also characterises classical and quantum phase transitions [6,17], dynamical quantum phase transitions [18,19], as well as phase transitions in steady states of dissipative dynamics [20–22], as it is proportional to the Bures metric in the state space (except for pathological, eliminable singularities [23,24]). Therefore, the QFI is expected to be much larger, i.e. superextensive, at critical points that separate macroscopically different phases, while it is at most extensive elsewhere. For certain topological [25] and non-equilibrium [20,21] phase transitions, the QFI can also be superex-

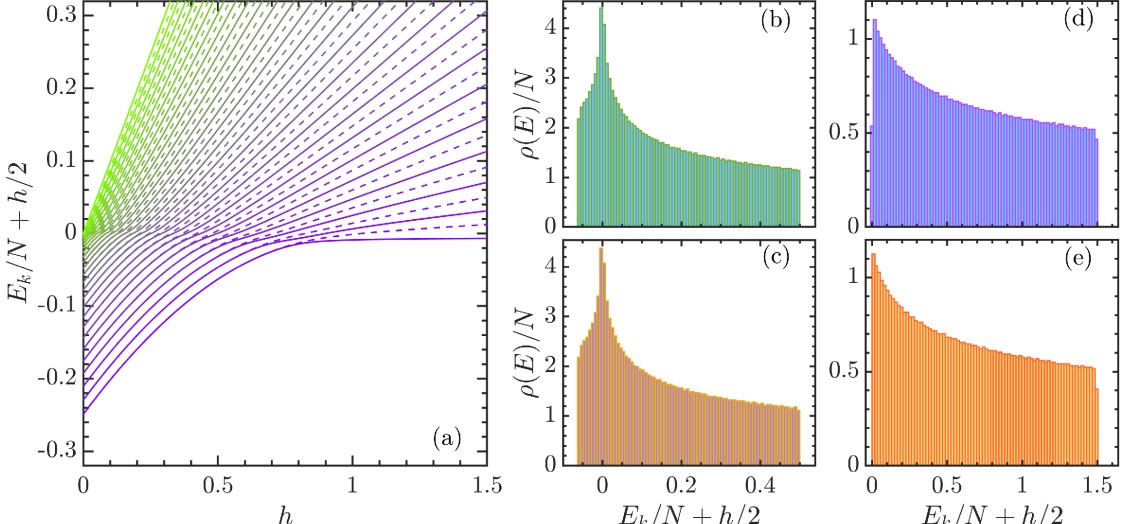

Figure 1: Panel (a): the rescaled energy levels of the Hamiltonian (1) as a function of the control parameter $h$ for $N = 50$. Solid lines represent the even-parity levels, while the dashed lines are the odd-parity levels: note the degeneracies at small $h$. Panels (b-c): rescaled density of eigenstates of the LMG Hamiltonian in the even-parity (b) and odd-parity (c) sectors with $h = 0.5$ and $N = 12000$. Panels (d-e): rescaled density of Hamiltonian eigenstates in the even-parity (b) and odd-parity (c) sectors with $h = 1.2$ and $N = 12000$.

tensive within an entire phase. All the aforementioned paradigms for phase transitions have then potential applications in precision metrology.

A class of phase transitions little studied under this approach [26, 27] consists of the so-called excited state quantum phase transitions (ESQPTs), identified by non-analiticities in the density of Hamiltonian eigenstates [28–32].

In this paper, we combine the two applications of the QFI: the characterisation of ESQPTs, and the proposal of efficient magnetometry schemes that greatly outperform the shot-noise estimations and even known protocols based on entangled states. The model under study is the so-called Lipkin-Meshkov-Glick (LMG) Hamiltonian which was originally conceived to test approximations in superconducting systems [33–35] and in nuclear systems [36–38]. More recently, the LMG model has been experimentally realised with molecular magnets [39, 40], Bosonic Josephson junctions in a spinor Bose-Einstein condensate [41, 42], and trapped ions [43–46]. Other proposals are based on Bose-Einstein condensates in an optical cavity [47] and cavity QED [48]. Implementations of our metrological schemes can benefit of the great level of control and stability reached by these experimental realisations, as already happened for matter wave interferometric phase estimations [49–51].

We introduce the model Hamiltonian and its properties in section 2. In section 3, the characterisation of the ESQPT in terms of the QFI peaks is discussed. Two concrete magnetometric protocols based on the system spectral properties induced by the ESQPT are described in section 4. We also analyse accuracy and running time scalings in order to compare the performances of these protocols with standard, shot-noise limited estimations and with metrological schemes using entangled states. The robustness of magnetometry against several noise sources is analysed in section 5. Conclusions are drawn in section 6, and some technical details are reported in appendices.

## 2 Model

The LMG Hamiltonian describes $N$ spins $1/2$ with long-range interaction and immersed in an external magnetic field $h$,

$$H_h = h\hat{S}_z - \frac{\hat{S}_x^2}{N}, \tag{1}$$

where $\hat{S}_\alpha = \sum_{i=1}^N \hat{\sigma}_\alpha^i/2$ ($\alpha = x, y, z$) are the collective spin operators, and $\hat{\sigma}_\alpha^i$ denotes the Pauli matrices of the $i$th spin. $H$ commutes with $\hat{S}^2 = \hat{S}_x^2 + \hat{S}_y^2 + \hat{S}_z^2$ whose eigenvalues are $s(s+1)$ with $s \in [0, N/2]$. We will consider only the sector $s = N/2$ with dimension $N+1$. Within each subspace with fixed $s$, the Hamiltonian commutes also with the parity $(-1)^{s+\hat{S}_z}$ [30,52] which has the two degenerate eigenvalues $+1$ (even parity) and $-1$ (odd parity). The subspaces with even and odd parity are then decoupled. These symmetries allow us to compute numerical exact diagonaliziation of the LMG Hamiltonian in the orthogonal subspaces with $s = N/2$ and even or odd parity. Denote by $\{|E_k\rangle, E_k\}_k$ the Hamiltonian eigenstates with even parity and the corresponding eigenenergies, while $\{|\tilde{E}_k\rangle, \tilde{E}_k\}_k$ form the Hamiltonian eigensystem in the odd-parity sector, and label the eigenergies such that $E_k \leq E_{k+1}$ and $\tilde{E}_k \leq \tilde{E}_{k+1}$. The following numerical analysis is limited to the even parity sector which has dimension $\lfloor 1+N/2 \rfloor$ (where $\lfloor x \rfloor$ is the greatest integer less than or equal to $x$). We will however show that the Hamiltonian eigenstates with odd parity alter neither the characterisation of the ESQPT nor the performances of the metrological schemes discussed in section 4. Moreover, we consider only positive magnetic fields $h \geq 0$ because the Hamiltonian is unitarily invariant under the transformation $h \longleftrightarrow -h$ that corresponds to spin-flip: $\hat{H}_h = \hat{U}^\dagger \hat{H}_{-h} \hat{U}$ with $\hat{U} = \otimes_{i=1}^N \hat{\sigma}_x^i$.

The ground state of the Hamiltonian (1) undergoes a second-order quantum phase transition when $h$ crosses the critical point $h_c = 1$ which separates the symmetric phase ($h > h_c$) and the broken-symmetry phase ($h < h_c$) [53,54]. The LMG model also exhibits a ESQPT for magnetic fields such that $h < h_c$ [28,32,52,55–59].

ESQPTs manifest themselves by non-analytic density of Hamiltonian eigenstate for the LMG model [28,52,57]. Indeed, from figure 1, we observe that the energy levels are concentrated around energies such that $E/N + h/2 \approx 0$. Therefore, for a given value $h < h_c$, the density of Hamiltonian eigenstates, $\rho(E) = \sum_n \delta(E - E_n)$, exhibits a sharp, cusplike peak around the so-called critical energy $E_c = -hN/2$, which diverges logarithmically in the thermodynamic limit, i.e.,

$$\rho(E \approx E_c) \approx -\frac{\ln(2|E-E_c|/N)}{2\pi\sqrt{h(1-h)}}, \tag{2}$$

whereas it shows a smooth behavior for $h > h_c$ [28,57].

The above spectral properties of the LMG model indicate that an ESQPT can be reached either by tuning the energy for a fixed control parameter $h < h_c$, or varying the control parameter for any fixed excited state. The ESQPT occurs at a different field value $h$ for each excited state: this critical value is $h_c^k = -2E_k/N$ for the eigenstate $|E_k\rangle$, such that $E_k = E_c$. Another remarkable feature of figure 1(a) is the change of curvature of $E_k$ at $h = h_c^k$. The Hellmann-Feynman theorem, $\partial_h E_k = \langle E_k | \partial_h \hat{H}_h | E_k \rangle = \langle E_k | \hat{S}_z | E_k \rangle$, implies that the derivative of total magnetization along the $z$ direction is the curvature of $E_k$: $\partial_h^2 E_k = \partial_h \langle E_k | \hat{S}_z | E_k \rangle$, see appendix E.

Return for a moment to the Hamiltonian eigensystem in the odd-parity sector. The eigenvalues $\tilde{E}_k$ have the same qualitative shape of those in the even-parity sector, with $E_k < \tilde{E}_k < E_{k+1}$ for $h > h_c^k$. If, on the other hand, $h \leq h_c^k$ (or equivalently $E_k \leq E_c$), then $\tilde{E}_k \approx E_k$ except for the highest energy level with even $N$ that corresponds only to an even eigenstate (see figure 1(a)).

Therefore, the even- and odd-parity sectors share the same spectral properties: compare for instance the density of eigenstates in figure 1(b-e)). Moreover, the eigenstates $|E_k\rangle$ and $|\tilde{E}_k\rangle$ have very similar decompositions in the eigenbasis of $\hat{S}_x$ [52]: $|E_k\rangle$ and $|\tilde{E}_k\rangle$ differ only by relative phases if $h \leqslant h_c^k$ (i.e., $E_k = \tilde{E}_k \leqslant E_c$), while the participation ratio varies smoothly with the eigenenergy and the magnetic field values if $h > h_c^k$ (i.e, $\tilde{E}_k > E_k > E_c$). These similarities between the even- and odd-parity sectors indicate that the following analysis extends to the odd-parity sector.

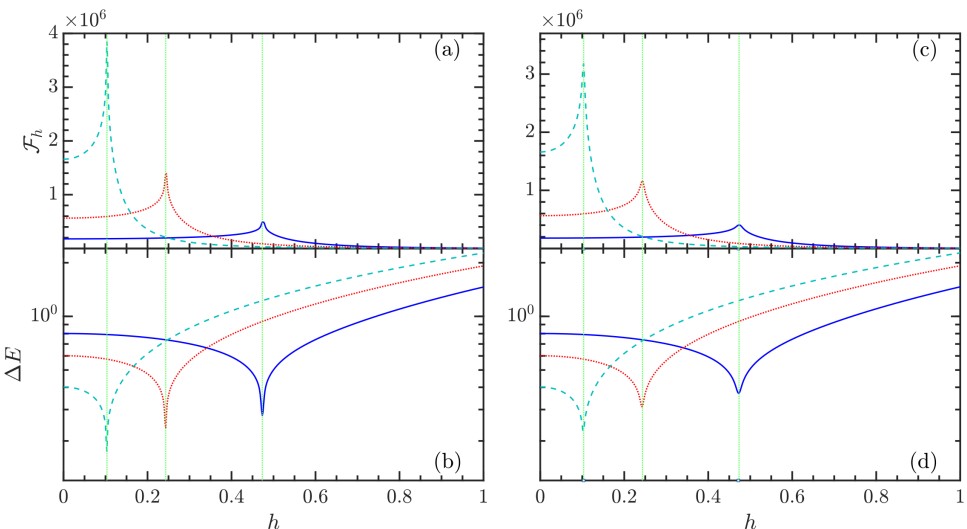

Figure 2: Panels (a,b): the QFI $\mathcal{F}_h(E_k)$ (a) and the eigenenergy gap $\Delta E = E_{k+1} - E_k$ (b) as a function of the control parameter $h$ for even-parity Hamiltonian eigenstates $|E_k\rangle$ with $k = 0.1N$ (blue solid lines), $k = 0.2N$ (red dotted lines), $k = 0.3N$ (cyan dashed lines) for $N = 800$. Panels (c,d): the QFI $\mathcal{F}_h(\tilde{E}_k)$ (c) and the eigenenergy gap $\Delta E = \tilde{E}_{k+1} - \tilde{E}_k$ (d) as a function of the control parameter $h$ for odd-parity Hamiltonian eigenstates $|\tilde{E}_k\rangle$ with $k = 0.1N$ (blue solid lines), $k = 0.2N$ (red dotted lines), $k = 0.3N$ (cyan dashed lines) for $N = 800$. The green vertical lines indicate the critical point for each eigenstate, $h_c^k = -2E_k/N$.

## 3   Excited state quantum phase transitions

We characterise the ESQPT in the LMG model through superextensive size scaling of the QFI. The latter is a function of the density matrix and of its variation with respect to a parameter that is the magnetic field $h$ in our case: see Appendix A for further details. The QFI $\mathcal{F}_h$ provides a lower bound for the variance of any unbiased estimation of the magnetic field using operations independent from $h$. This bound is known as the quantum Cramér-Rao bound [1, 2, 4]: $\delta^2 h \geqslant \left(M\mathcal{F}_h\right)^{-1}$, where $M$ is the number of independent measurements.

The QFI of the Hamiltonian eigenstate $|E_k\rangle$ with even parity and with eigenenergy $E_k$ reads [25, 60–63]

$$\mathcal{F}_h(E_k) = 4 \sum_{n \neq k} \frac{|\langle E_n|\partial_h \hat{H}_h|E_k\rangle|^2}{(E_n - E_k)^2} = 4 \sum_{n \neq k} \frac{|\langle E_n|\hat{S}_z|E_k\rangle|^2}{(E_n - E_k)^2} \simeq 4 \fint dE \, \rho(E) \frac{|\langle E|\hat{S}_z|E_k\rangle|^2}{(E - E_k)^2}, \quad (3)$$

where $\fint$ stands for the Cauchy principal value. The sum in equation (3) runs over all energy eigenstates, with both even and odd parity, but the odd-parity eigenstates $|\tilde{E}_n\rangle$ do not con-

tribute because $\hat{S}_z$ preserves the parity. Similarly, the QFI of odd-parity Hamiltonian eigenstates $|\tilde{E}_k\rangle$, denoted by $\mathcal{F}_h(\tilde{E}_k)$, is given by equation (3) with $\{|E_k\rangle, E_k\}_k$ replaced by $\{|\tilde{E}_k\rangle, \tilde{E}_k\}_k$. The QFI diverges if $E_n - E_k \rightarrow 0$, and thus the eigenenergy accumulation around $E_c$, due to the ESQPT, strongly influences the behaviour of the QFI.

The QFI and the energy gap $\Delta E$ between adjacent eigenstates as a function of $h$ are plotted in figure 2 for different Hamiltonian eigenstates. The peak in QFI corresponds to the minimum value of $\Delta E$ for each eigenstate. The QFI at different particle numbers has similar shapes with $N$-dependent heights. In order to study the $N$-dependence of the QFI, we fit the maximum value of QFI, namely $\mathcal{F}_{h,m}(E_k) = \max_h \mathcal{F}_h(E_k)$ for each eigenstate, its width at half of the peak, denoted by $\Sigma_h(E_k)$, and the local minimum value of the QFI at $h = 0$ with powers of $N$. Note that eigenstates with even and odd parity, $|E_k\rangle$ and $|\tilde{E}_k\rangle$, show very similar peaks at the same value of $h$ in figure 2, indicating that the aforementioned size scalings are similar for both parity sectors.

The maximum value of QFI, $\mathcal{F}_{h,m}(E_k)$, as a function of $N$ is plotted in figure 3(a), and fitted with $\mathcal{F}_{h,m}(E_k) = CN^\gamma$. Remarkably, the values of the exponent for each excited state are very close to each other: $\gamma \simeq 2.07$. The values of coefficients $C$ of this and of all the following power law fits are drawn in the Appendix B. The QFI local minimum $\mathcal{F}_{h=0}(E_k)$ is fitted by $CN^\delta$, with $\delta \simeq 2.01$ irrespective of the excited states, as shown in figure 3(b). The superextensivity of $\mathcal{F}_{h=0}(E_k)$ for all excited states is due to the eigenstate density $\rho(E)$ that remains moderately large compared to its peak, at energies smaller than the critical one $E_c$ (see figure 1(b,c) and [52,57]).

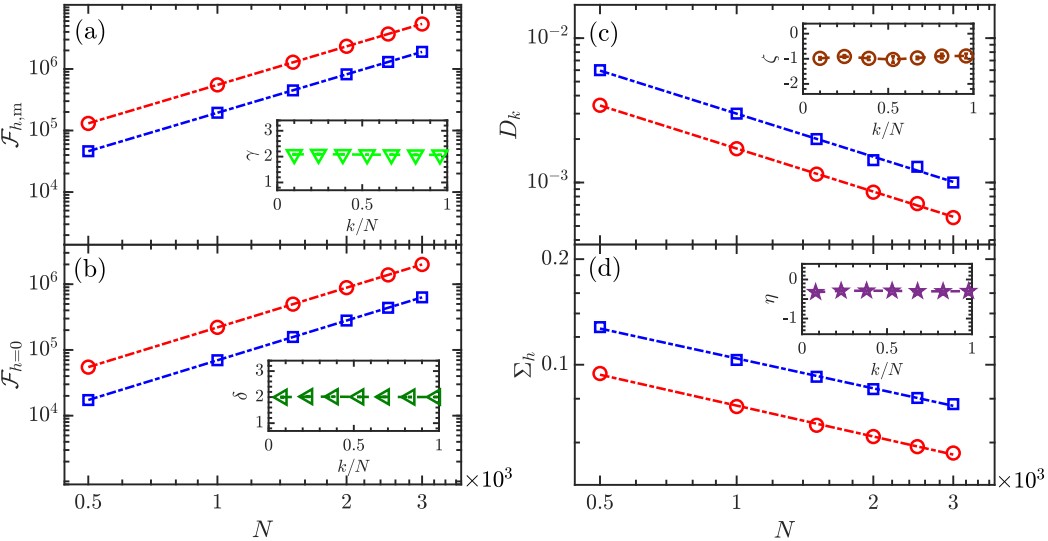

Figure 3: The maximum value of the QFI $\mathcal{F}_{h,m}(E_k)$ (panel (a)), $\mathcal{F}_{h=0}(E_k)$ (panel (b)), $D_k$ (panel (c)), and $\Sigma_h(E_k)$ (panel (d)) as a function of the system size $N$ in log-log scale. For all of panels, blue squares represent values for the Hamiltonian eigenstate $|E_k\rangle$ with $k = 0.1N$, and red circles correspond to $k = 0.2N$. The insets show plots of exponents of power law fits of the corresponding panels for several Hamiltonian eigenstates.

We further compare the QFI width $\Sigma_h(E_k)$ with the absolute difference, $D_k = |h_c^{k+1} - h_c^k|$, between the critical fields respectively for the $(k + 1)$-th and the $k$-th excited states at finite size. Panels (c) and (d) of figure 3 show the fits $D_k = CN^\zeta$ and $\Sigma_h(E_k) = CN^\eta$, where the exponents $\zeta \simeq -1.03$ and $\eta \simeq -0.299$ are insensitive to the eigenstates. The QFI width $\Sigma_h$ is significantly larger than the spacing $D_k$, because of the logarithmic divergence of the eigenstate

density $\rho(E)$ (see figure 1(b,c)). Therefore, signatures or forerunners of the ESQPT are found in a neighbourhood of the corresponding critical excited state.

In order to strengthen the characterisation of the ESQPT via the QFI, we plot $\mathcal{F}_h(E_k)$ as a function of the eigenenergy and at fixed control parameters $h$ in figure 4, as the ESQPT can be driven by tuning the energy. $\mathcal{F}_h$ exhibits a sharp peak close to the critical energy $E_c$, which is a signature of the ESQPT. The maximum value $\mathcal{F}_{h,m}^* = \max_{E_k} \mathcal{F}_h(E_k)$ is plotted in figure 4(c) for different control parameters, and fitted by $\mathcal{F}_{h,m}^* = CN^\xi$ where the exponent is $\xi \simeq 2.07$, almost insensitive to the control parameter $h$. Note that $\xi \simeq \gamma$ so we assume these exponents are equal, as expected since they both describe the size scaling of the QFI at critical points. The width at half peak of the QFI expressed as a function of the rescaled eigenenergy $E_k/N$ in figure 4(d) also shows a power scaling with the system size $N$, i.e. $\Sigma_E^*(h) = CN^\mu$ where the exponent $\mu \simeq -0.227$ is almost independent of the control parameter $h$.

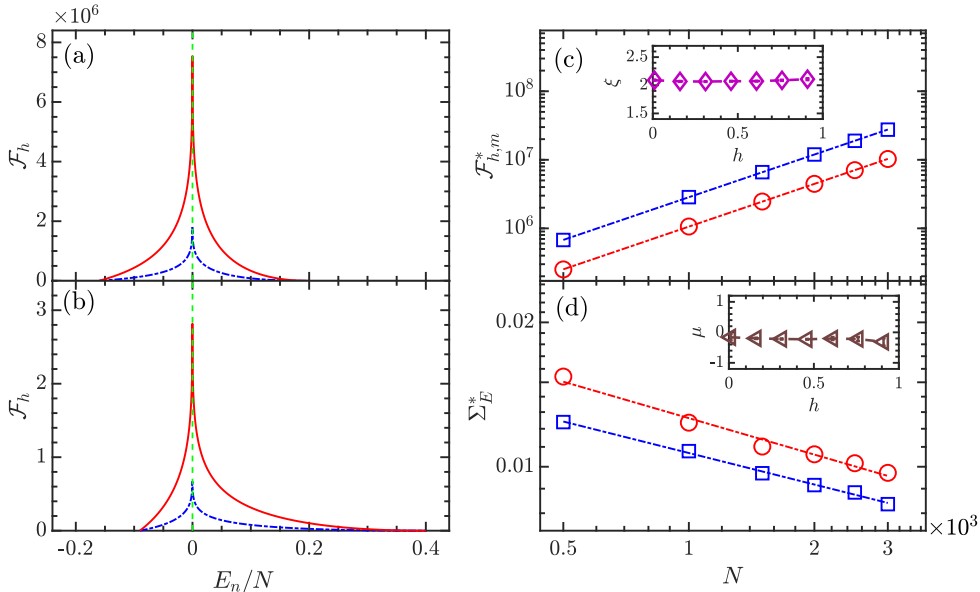

Figure 4: The QFI, $\mathcal{F}_h(E_n)$, as a function of rescaled eigenenergies, $E_n/N$, for $N = 800$ (blue dot-dashed lines) and $N = 1600$ (red solid lines) with $h = 0.2$ (panel (a)) and $h = 0.4$ (panel (b)). The vertical dotted lines indicate the critical energy $E_c = -hN/2$ of the ESQPT. Panels (c) and (d) are respectively the maximum values, $\mathcal{F}_{h,m}^*$, and the widths, $\Sigma_E^*(h)$, of the above plots as a function of the system size $N$ in log-log scale for $h = 0.2$ (blue squares) and $h = 0.4$ (red circles); exponents of power law fits of the plots for several control parameters $h$ are shown in the insets.

From the superextensivity of the QFI around the magnetic fields $h_c^k$ and the energy $E_c$, we now estimate the scaling of energy gaps $|E_n - E_k|$ around the value $E_c$. The QFI (3) scales as $N^\gamma$ for eigenstates with energies close to $E_c$ within a width $N\Sigma_E^*(h) \sim N^{1+\mu}$, where the factor $N$ is due to the energy rescaling, $E_k/N$ in figure 4. The superextensive scaling of the QFI and $\langle E_n|\hat{S}_z|E_k\rangle = \mathcal{O}(N)$ imply that $E_n - E_k$ in equation (3) must scale subextensively around $E_c$. In order to obtain an estimation for $E_n - E_k$, let us use the scaling of the eigenstate density (2) around $E_c$, $\rho(E \approx E_c) \sim \ln N$ in equation (3). A power law size scaling for $\mathcal{F}_{h,m}$ is consistently recovered by using the ansatz $(E_n - E_k)^2 = \mathcal{O}(N^{2-\nu}\ln N)$ with $\nu > 0$. We then estimate

$$\mathcal{F}_h(E_k) \sim 4 \oint_{E_c - \frac{1}{2}N\Sigma_E^*(h)}^{E_c + \frac{1}{2}N\Sigma_E^*(h)} dE\, \rho(E) \frac{|\langle E|\hat{S}_z|E_k\rangle|^2}{(E - E_k)^2} \sim N^{1+\nu}\Sigma_E^*(h) \sim N^{1+\mu+\nu}. \tag{4}$$

Comparing this estimate with the scaling $\mathcal{F}_h(E_k) \sim N^\gamma$, we obtain $\nu = \gamma - \mu - 1 \simeq 1.3$.

# 4 Magnetometric protocols

The metrological interpretation of the Fisher information and the results in the previous section imply that systems prepared in nearly critical excited states of $H_h$ are good probes for the estimation of $h$ ranging in $[-h_c, h_c]$[1] with the smallest achievable variance $\mathcal{F}_{h,m}^{-1} \sim N^{-2.07}$. In the following, we discuss two concrete magnetometric protocols.

## 4.1 Protocol 1

The first protocol is divided in two steps, namely preparation of the probe state and measurement of observables from which the magnetic field is inferred. We then analyse the time overhead of the protocol.

### 4.1.1 Probe preparation

The probe preparation of the first protocol consists in sampling Hamiltonian eigenstates. One starts with preparing any state with significant overlap with the eigenstates having energies around $E_c$. An example is the pure state with all spins pointing down in the $z$ direction, $|\downarrow_z\rangle^{\otimes N}$, whose overlap with the critical eigenstate at any $h$ is much larger than the overlap with the other eigenstates [52]. Alternatively, one can consider a state whose support is the entire subspace with $s = N/2$: the Gibbs state restricted to this subspace is an example, and can be prepared when the systems interacts with a thermal bath [64, 65] (see Appendix C). The restriction to the subspace with $s = N/2$ is realized with dynamics conserving the total spin of the system, in order to ensure that the system state never leaves the desired subspace. This condition is met for system-bath interactions that are symmetric under spin permutation and with symmetric initial states. The restriction to the subspace with $s = N/2$ can also be realized by dispersive vibrational sideband excitation for trapped ions [66, 67], or exploiting the permutation symmetry implied by particle inditinguishability, e.g. for ultracold Bosons used to implement the LMG model [41, 42].

After the preparation of the above states, e.g. $|\downarrow_z\rangle^{\otimes N}$ or the Gibbs state in the subspace with $s = N/2$, a projective measurement on eigenstates of the unitary time-evolution $\hat{U}_{\Delta t} = e^{-i\hat{H}_h \Delta t}$ is realized by the phase estimation algorithm [68, 69]. The eigenstates of $\hat{U}_{\Delta t}$ are also eigenstates of $\hat{H}_h$, and thus the phase estimation algorithm provides an effective measurement of energy. This algorithm requires $\tilde{d}$ ancillary qubits prepared in the state $(|0\rangle + |1\rangle)^{\otimes \tilde{d}}/2^{\tilde{d}/2}$, the implementation of controlled unitaries $\hat{C}_U(j)$ ($j = 0, 1, \ldots, \tilde{d} - 1$) where the control qubits are the ancillary qubits, the inverse quantum Fourier transform of the ancillary qubits and their measurement in the computational basis (eigenbasis of $\sigma_z$). The controlled unitaries $\hat{C}_U(j)$ are unitary operations $\hat{U}_{2^j \Delta t} = e^{-i\hat{H}_h 2^j \Delta t}$ applied to the system state depending on the state of a control particle: $\hat{C}_U(j)|0\rangle \otimes |\psi\rangle = |0\rangle \otimes |\psi\rangle$ and $\hat{C}_U(j)|1\rangle \otimes |\psi\rangle = |1\rangle \otimes \hat{U}_{2^j \Delta t}|\psi\rangle$, where $|\psi\rangle$ is any system pure state, and the other state pertains the control qubit. The controlled unitaries $\hat{C}_U(j)$ can be realised although the Hamiltonian $\hat{H}_h$ is unknown, reducing it to the time-evolution $\hat{U}_{2^j \Delta t}$, with several schemes [70–74] some of which tailored for and experimentally realised with photons and trapped ions.

Just before the measurements of the ancillary qubits, these qubits are entangled with the system. After the measurements, the system state collapses in one the Hamiltonian eigenstates $\{|E_k\rangle, |\tilde{E}_k\rangle\}_k$, and the measurement outcomes are the binary representation of $2^{\tilde{d}}(E_k \Delta t \mod 2\pi)$ within an error $\pi$ and with probability at least $4/\pi^2 \simeq 0.405$. The error for estimating the phase $\phi_k = E_k \Delta t \mod 2\pi$ is then $\pi/2^{\tilde{d}}$. The success probability increases to $p_{\text{succ}}$ if $\phi_k$

---

[1]Remind that the Hamiltonian for negative $h$ is unitarily equivalent to that for positive $h$, and thus our results generalise to negative magnetic fields.

is estimated using only the first $d = \tilde{d} - \left\lceil \log_2 \left( \frac{2-p_{\text{succ}}}{2-2p_{\text{succ}}} \right) \right\rceil$ ancillary qubits, while the error of this estimation becomes $\epsilon = \pi/2^d$. It is important to notice that only a small number of ancillary qubits are used to reach a finite but high success probability $p_{\text{succ}}$, namely $\left\lceil \log_2 \left( \frac{2-p_{\text{succ}}}{2-2p_{\text{succ}}} \right) \right\rceil$, and therefore $d = \mathcal{O}(\tilde{d})$. For instance, $p_{\text{succ}} = 0.9$ implies $\left\lceil \log_2 \left( \frac{2-p_{\text{succ}}}{2-2p_{\text{succ}}} \right) \right\rceil = 3$. We address general probabilistic performances of our metrological problem in section 4.3.

From the above discussion, the phase estimation algorithm can be used to simultaneously sample the eigenstates $\{|E_k\rangle, |\tilde{E}_k\rangle\}_k$ and to measure the corresponding phases $\{\phi_k\}_k$. Afterwards, one picks up the sampled state with energy closest to critical energy $E_c$, that is the state with the largest sampled phase density which inherits the scaling in equation (2) (see Appendix D). When the system state before the phase estimation algorithm is $|\downarrow_z\rangle^{\otimes N}$, then $\hat{S}_z|\downarrow_z\rangle^{\otimes N} = -\frac{N}{2}|\downarrow_z\rangle^{\otimes N}$, and $|\downarrow_z\rangle^{\otimes N}$ belongs to the even parity sector. Consequently, only even-parity Hamiltonian eigenstates, $|E_k\rangle$, result from the probe preparation. This is not the case for the Gibbs state that has support on the whole sector with $s = N/2$: the state after probe preparation is the mixture $\varrho_k = \frac{1}{2}\left(|E_k\rangle\langle E_k| + |\tilde{E}_k\rangle\langle \tilde{E}_k|\right)$ if $h \leq h_c^k$ (i.e., $E_k \leq E_c$), while it is one of the pure states $|E_k\rangle$ and $|\tilde{E}_k\rangle$, with probabilities $e^{-\beta E_k}/\left(\sum_k e^{-\beta E_k} + \sum_k e^{-\beta \tilde{E}_k}\right)$ and $e^{-\beta E_k}/\left(\sum_k e^{-\beta \tilde{E}_k} + \sum_k e^{-\beta E_k}\right)$ respectively, for $h > h_c^k$ (i.e., $E_k > E_c$).

### 4.1.2 Measurement of magnetisation

Consider that the probe state prepared as described above is the even-parity eigenstate $|E_k\rangle$ with $E_k \approx E_c$. This is the only relevant case when the state before the phase estimation algorithm is $|\downarrow_z\rangle^{\otimes N}$, as mentioned in the previous subsection. We return soon to the odd-parity sector. The estimation of $h$ is inferred from a measurement of the magnetisation $\langle E_k|\hat{S}_z|E_k\rangle$, and from its theoretical dependence from $h$. Denoting the magnetisation variance by

$$\Delta^2_{|E_k\rangle}S_z = \langle E_k|\hat{S}_z^2|E_k\rangle - \langle E_k|\hat{S}_z|E_k\rangle^2$$

and applying error propagation, the estimation accuracy is

$$\delta^2 h = \frac{\Delta^2_{|E_k\rangle}S_z}{\left(\partial_h \langle E_k|\hat{S}_z|E_k\rangle\right)^2} \ . \tag{5}$$

The total magnetization along the $z$ axis, $\langle E_k|\hat{S}_z|E_k\rangle$, show minima with large derivatives at the critical points of the ESQPT, as shown in figure 5(a). The fits of the estimation error (5) with $\delta^2 h = CN^{\chi_\pm}$ for the magnetic field approaching the critical fields $h \to h_c^{k\pm}$ from smaller (minus sign) or larger (plus sign) values are plotted in figures 5(c,d). The exponents are $\chi_+ \simeq -1.51$ and $\chi_- \simeq -1.44$ on average over the Hamiltonian eigenstates, and entail an accuracy for the magnetic field (5) better than standard shot-noise $\mathcal{O}(1/N)$.

The above discussion already entails that the magnetometric protocol 1 can be realized with high accuracy, given the initial state $|\downarrow_z\rangle^{\otimes N}$. For completeness, we also show that the estimation accuracy has the same scaling with $N$ when the system in initially the Gibbs state. Consider, first, probes prepared in the mixed state $\varrho_k = \frac{1}{2}\left(|E_k\rangle\langle E_k| + |\tilde{E}_k\rangle\langle \tilde{E}_k|\right)$ that happen when $h \leq h_c^k$ (see subsection 4.1.1). The numerator and the denominator of the estimation accuracy (5) are then replaced respectively by

$$\Delta^2_{\varrho_k}S_z = \text{Tr}(\varrho_k \hat{S}_z^2) - \text{Tr}(\varrho_k \hat{S}_z)^2 = \frac{1}{2}\left(\langle E_k|\hat{S}_z^2|E_k\rangle + \langle \tilde{E}_k|\hat{S}_z^2|\tilde{E}_k\rangle\right) - \frac{1}{4}\left(\langle E_k|\hat{S}_z|E_k\rangle + \langle \tilde{E}_k|\hat{S}_z|\tilde{E}_k\rangle\right)^2 , \tag{6}$$

$$\left(\partial_h \text{Tr}(\varrho_k \hat{S}_z)\right)^2 = \frac{1}{4}\left(\partial_h \langle E_k|\hat{S}_z|E_k\rangle + \partial_h \langle \tilde{E}_k|\hat{S}_z|\tilde{E}_k\rangle\right)^2 , \tag{7}$$

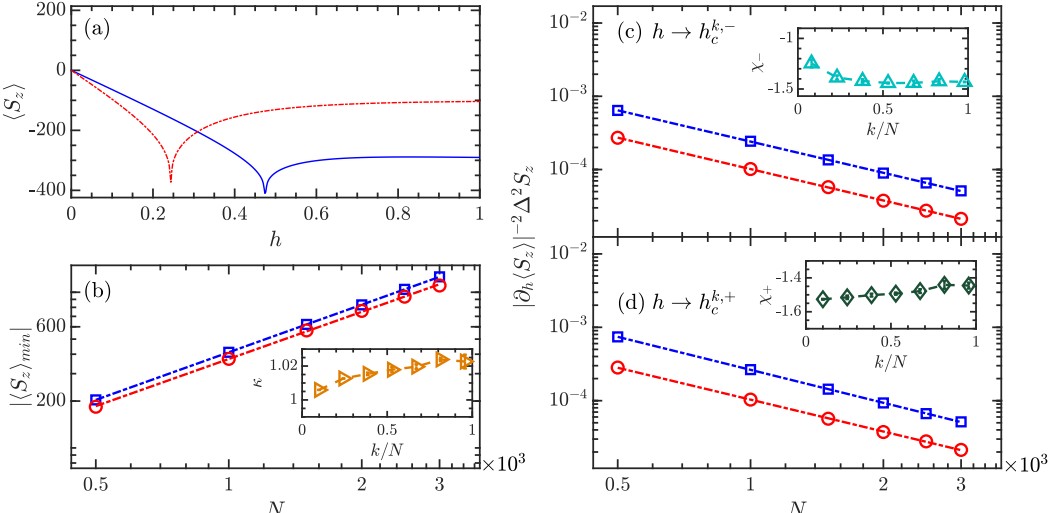

Figure 5: Panel (a) is the magnetization $\langle \hat{S}_z \rangle$ as a function of $h$ for different Hamiltonian eigenstates and $N = 1000$. Panel (b) is the log-log plot of the modulus of the minimum of $\langle S_z \rangle$ as a function of $N$. Panels (c) and (d) are log-log plots of $\left| \partial_h \langle \hat{S}_z \rangle \right|^{-2} \Delta^2 S_z$ as a function of the system size $N$ at the corresponding critical fields for $h \to h_c^{k-}$ and $h \to h_c^{k+}$ respectively. For all of panels, blue squares represent values for the Hamiltonian eigenstate $|E_k\rangle$ with $k = 0.1N$, and red circles correspond to $k = 0.2N$. The insets show plots of exponents of power law fits of the corresponding panels for several Hamiltonian eigenstates.

where we have used the fact that $\hat{S}_z$ preserves the parity, such that $\langle E_k | \hat{S}_z | \tilde{E}_k \rangle = \langle E_k | \hat{S}_z^2 | \tilde{E}_k \rangle = 0$. The energy degeneracy $E_k = \tilde{E}_k$ for $h \leqslant h_c^k$ and the Hellmann-Feynman theorem, i.e. $\partial_h E_k = \langle E_k | \hat{S}_z | E_k \rangle$ and $\partial_h \tilde{E}_k = \langle \tilde{E}_k | \hat{S}_z | \tilde{E}_k \rangle$, imply $\langle E_k | \hat{S}_z | E_k \rangle = \langle \tilde{E}_k | \hat{S}_z | \tilde{E}_k \rangle$ and $\partial_h \langle E_k | \hat{S}_z | E_k \rangle = \partial_h \langle \tilde{E}_k | \hat{S}_z | \tilde{E}_k \rangle$ (see appendix E). These equalities allow us to simplify equations (6) and (7), and to write the estimation accuracy, given the probe state $\varrho_k$ and $h \leqslant h_c^k$, as

$$\delta^2 h = \frac{\Delta^2_{\varrho_k} S_z}{\left( \partial_h \mathrm{Tr}(\varrho_k \hat{S}_z) \right)^2} = \frac{\Delta^2_{|E_k\rangle} S_z}{2 \left( \partial_h \langle E_k | \hat{S}_z | E_k \rangle \right)^2} + \frac{\Delta^2_{|\tilde{E}_k\rangle} S_z}{2 \left( \partial_h \langle \tilde{E}_k | \hat{S}_z | \tilde{E}_k \rangle \right)^2} \ . \tag{8}$$

The magnetisation is an extensive quantity, and thus $\langle E_k | S_z | E_k \rangle = \mathcal{O}(N)$. We have numerically checked this scaling for the minima of $\langle E_k | S_z | E_k \rangle$ attained at the critical points $E_k = \tilde{E}_k = E_c$ and $h = h_c^k$ (see figure 5(a)). The minimum values are fitted by $|\langle \hat{S}_z \rangle_{\min}| = C N^\kappa$, resulting in $\kappa \simeq 1.02$ irrespective of the excited state $|E_k\rangle$ (see figure 5(b)), where the deviation of $\kappa$ from 1 is due to numerical errors. The extensivity of the magnetisation also entails that $\langle E_k | S_z^2 | E_k \rangle$ and $\langle \tilde{E}_k | S_z^2 | \tilde{E}_k \rangle$ scale as $\mathcal{O}(N^2)$. Moreover, the Hellmann-Feynman theorem and the structure of Hamiltonian eigenstates in the $\hat{S}_x$ eigenbasis [52] imply that the two terms in the estimation accuracy (8) share the same scaling with $N$ (see appendix E for further details): the first term is half of equation (5) discussed above and plotted in figures 5(c,d).

If the probe is prepared in the eigenstate $|\tilde{E}_k\rangle$, namely for an initial Gibbs state and $h \geqslant h_c^k$, the estimation accuracy is as in equation (5) with $|E_k\rangle$ replaced by $|\tilde{E}_k\rangle$. Arguments similar to the above ones (see also appendix E), can still be used to show that the estimation accuracy has the same $N$ scaling as if even energy eigenstates were considered, and in particular the same scaling then the above cases when the magnetic field approaches $h_c^k$. The difference is that instead of using eigenvalue equalities we use bounds $E_k \leqslant \tilde{E}_k \leqslant E_{k+1}$, $\partial_h E_k \leqslant \partial_h \tilde{E}_k \leqslant \partial_h E_{k+1}$,

and $\partial_h^2 E_k \leqslant \partial_h^2 \tilde{E}_k \leqslant \partial_h^2 E_{k+1}$ shown in figure 1(a) for $h > h_c^k$. Note, for instance, that the change of curvature of $\tilde{E}_k$ at $h = h_c^k$ implies that $\langle \tilde{E}_k | \hat{S}_z | \tilde{E}_k \rangle$ exhibits a cusp-like minimum at $h = h_c^k$ with the same minumum value as in figure 5(a) since $\langle \tilde{E}_k | S_z | \tilde{E}_k \rangle$ approaches $\langle E_k | S_z | E_k \rangle$ as $h \to h_c^k$ from larger values.

### 4.1.3 Time overhead

We now estimate that the running time of protocol 1 is constant in $N$. Therefore, sub-shot-noise accuracy is achieved without time overhead with respect to standard, shot-noise limited estimations.

The main source for time overhead is the phase estimation algorithm. Indeed, the magnetisation measurement is implemented in time $\mathcal{O}(N^0)$ by measuring $\sigma_z^i$ for all $i = 1, \dots, N$ simultaneously. Also the realisation of the initial state for the probe preparation step is implemented in constant time. An example is the pure state with all spins down in the $z$ direction that is realised by preparing all the spins simultaneously. The other example is the Gibbs state, which can be realised as the steady state of Markovian dynamics (see appendix C) that typically shows exponential time decay $e^{-t/\tau}$, where the relaxation time $\tau$ decreases by increasing the system-bath coupling constant $\lambda$. A specific thermalizing dynamics for the LMG model with $\tau = \mathcal{O}(1/\lambda)$ and constant in $N$ has been studied in [75].

We stress however that the full relaxation is not needed as it is enough to prepare a state with support on the entire subspace with $s = N/2$, and this condition is typically met even at smaller times. Alternative states are Gibbs states with any Hamiltonian which commutes with $S^2$, such that the restriction to the subspace $s = N/2$ can be implemented as described above, and that exhibits fast thermalization. There are many of such Hamiltonians, e.g. non-interacting spin Hamiltonian ($\tau = \mathcal{O}(1/\lambda)$) [76] coupled to blackbody radiation or the Hamiltonian (1) with the addition of a term $-\frac{1}{N}\hat{S}_y^2$ and with a magnetic field larger than $h_c$ ($\tau$ decreasing with increasing $N$) [77].

We now focus on the running time of the phase estimation algorithm which measures the phases of $\hat{U}_{\Delta t}$, that are $\phi_k = E_k \Delta t \mod 2\pi$, with accuracy $\epsilon = \pi/2^d \leqslant \mathcal{O}(N^0)$, using $d$ ancillary qubits. The algorithm uses a number $\mathcal{O}(\tilde{d}^2) = \mathcal{O}(d^2) = \mathcal{O}(\log 1/\epsilon)^2$ of single- and two-spin operations and controlled unitaries $\hat{C}_U(j)$ with $j = 0, 1, \dots, d-1$. The density of the measured phases $\phi_k$ is different from the eigenenergy density due to the periodicity modulo $2\pi$: indeed, two energy eigenvalues can be confused, unless $\Delta t \leqslant 2\pi/\max_{k,k'}\{|E_k - E_{k'}|\}$ $= \mathcal{O}(1/N)$. So, the running time is $\mathcal{O}(\log 1/\epsilon)^2 + \mathcal{O}(\epsilon N)^{-1}$ constant in $N$ for large $N$. The condition on the time scale $\Delta t$ can be relaxed at the cost of errors occurring with small probability (see Appendix D). If $\Delta t = \mathcal{O}(N^0)$, the running time is again constant in $N$, namely $\mathcal{O}(\log 1/\epsilon)^2 + \mathcal{O}(1/\epsilon)$. This proves that the phase estimation algorithm in our protocol is much simpler than in quantum computation where the running time grows with the qubit number.

Several experimental efforts has been recently made to realise scalable phase estimation algorithms [78–83]. Moreover, the number of ancillary qubits and that of operations in the phase estimation algorithm can be reduced by infering multiple phase digits at a time [84], by employing time series from the expectations of $\hat{U}_t$ [85] or from an adaptive scheme based on circular statistics [86].

### 4.2 Protocol 2

We now discuss a second protocol entirely based on the phase estimation algorithm. The protocol starts with the same operations as in the probe preparation of protocol 1. The aim of this part is not the preparation of the critical eigenstate as in the previous protocol, but rather the measurement of the phase $\phi_c^{(N)} = E_c^{(N)}\Delta t \mod 2\pi$ corresponding to the critical energy

$E_c^{(N)}$. We have stressed in the notation that the above quantities correspond to a system made of $N$ particles.

It is in general hard to infer $E_c$ from $\phi_c$, because one does not know how many times $E_c^{(N)}\Delta t = -Nh\Delta t/2$ wraps up the interval $[0, 2\pi)$ for unknown fields $h$. For the same reason, the theoretical dependence of $\phi_c$ from the field $h$ is not known in general. Therefore, it is not possible to estimate $h$ from a measurement of the critical eigenenergy $E_k \xrightarrow[N\gg 1]{} E_c$. In order to overcome this difficulty, one can run the quantum estimation algorithm with a system with $N + 1$ particles, measure $\phi_c^{(N+1)} = E_c^{(N+1)}\Delta t \mod 2\pi$, and then compute the difference $\Delta\phi_c = \phi_c^{(N)} - \phi_c^{(N+1)}$. Indeed, there exist $m^{(N)}, m^{(N+1)} \in \mathbf{Z}$ such that $\phi_c^{(N)} = E_c^{(N)}\Delta t + 2m^{(N)}\pi \in [0, 2\pi)$ and $\phi_c^{(N+1)} = E_c^{(N+1)}\Delta t + 2m^{(N+1)}\pi \in [0, 2\pi)$. Choosing moreover $\Delta t$ such that $\left|E_c^{(N)} - E_c^{(N+1)}\right|\Delta t = |h|\Delta t/2 \leqslant \pi/2$ (e.g. $\Delta t \leqslant \pi$ for $|h| \leqslant h_c = 1$), we have three cases:

i)  $m^{(N)} = m^{(N+1)}$ then $\Delta\phi_c = \left(E_c^{(N)} - E_c^{(N+1)}\right)\Delta t = h\Delta t/2$ and $|\Delta\phi_c| \leqslant \pi/2$;

ii) $m^{(N+1)} = m^{(N)} + 1$ occurs only if $h > 0$ since $E_c^{(N)} > E_c^{(N+1)}$ in this case, then
$\Delta\phi_c = \left(E_c^{(N)} - E_c^{(N+1)}\right)\Delta t - 2\pi = h\Delta t/2 - 2\pi$ and $-2\pi < \Delta\phi_c < -3\pi/2$;

ii) $m^{(N+1)} = m^{(N)} - 1$ occurs only if $h < 0$ since $E_c^{(N)} < E_c^{(N+1)}$ in this case, then
$\Delta\phi_c = \left(E_c^{(N)} - E_c^{(N+1)}\right)\Delta t + 2\pi = h\Delta t/2 + 2\pi$ and $3\pi/2 < \Delta\phi_c < 2\pi$.

The three cases above are distinguished by $\Delta\phi_c$ lying in well separated intervals, and therefore it is always clear which formula for $\Delta\phi_c$ one has to consider for the estimation of $h$ even when small errors occur in the determination of $\Delta\phi_c$.

The value of the field $h$ can be estimated from $\Delta\phi_c$ which is measured with the accuracy $2\epsilon = 2\pi/2^d$ using the phase estimation algorithm. The accuracy $2\epsilon$ is doubled with respect to that of the phase estimation algorithm in protocol 1, because the measurement errors of two phases contribute to the error propagation formula for $\Delta\phi_c$. The error propagation leads to the following accuracy for $h$:

$$\delta^2 h = \frac{4\epsilon^2}{\left(\partial_h\Delta\phi_c\right)^2} = \frac{\pi^2}{4^{d-2}(\Delta t)^2}\,. \tag{9}$$

At finite size, $\partial_h\Delta\phi_c$ is replaced by

$$\partial_h\Delta\phi_k = \left(\partial_h E_k^{(N)} - \partial_h E_k^{(N+1)}\right)\Delta t \tag{10}$$

with $E_k \xrightarrow[N\gg 1]{} E_c$. Furthermore, the Hellmann-Feynman theorem (see appendix E) implies $\partial_h E_k = \langle E_k|\partial_h\hat{H}_h|E_k\rangle = \langle E_k|\hat{S}_z|E_k\rangle$, which scales as $\mathcal{O}(N)$ as discussed in section 4.1.2 (see figure 5(b)). This scaling numerically confirms that $\partial_h\Delta\phi_k$ is finite and constant in $N$: indeed, from equation (10), $\partial_h\Delta\phi_k \propto (N - (N + 1))\Delta t$.

Remarkably, the eigenenergies of both the even- and the odd-parity sectors accumulate around $E_c$, as discussed in section 2. Therefore, the measured $\Delta\phi_c$ and the magnetic field estimation accuracy (9) do not depend on the parity of the prepared probe states. Moreover, $\langle E_k|\hat{S}_z|E_k\rangle = \langle\tilde{E}_k|\hat{S}_z|\tilde{E}_k\rangle$ for $h \leqslant h_c^k$, such that the derivative (10) at finite size do not depend on the parity of the probe state as well.

The accuracy (9) can beat the shot-noise limit $\mathcal{O}(1/N)$ and indeed any polynomial scaling $\mathcal{O}(1/N^\theta)$ with logarithmically many ancillary qubits $d > \mathcal{O}(\theta\log_2 N)$. Protocol 2 can therefore beat the scaling provided by the inverse of the QFI that is $1/N^\gamma$. This is possible by employing measurements that depend on the parameter to be estimated $h$ (in our case through the time-evolution $\hat{U}_t = e^{-i\hat{H}_h t}$), similarly, for instance, to the use of energy measurements for Hamiltonian parameter estimations [87–90].

Although equation (9) does not depend on the specific system Hamiltonian (1), such accuracy can be achieved only if one identifies the critical eigenstate and the corresponding phase $\phi_c$. This aim is accomplished, as in section 4.1.1, by picking up the outcome of the phase estimation algorithm with higher density which scales as (2). The peculiar spectral properties provided by the ESQPT are therefore crucial, while identifying the resulting eigenstate is much harder for Hamiltonians without ESQPTs.

### 4.2.1 Time overhead

We now estimate the time overhead of protocol 2 and show that employing this time to repeat standard estimations, each with shot-noise (or even higher) accuracy, do not overcome the protocol performances. Assume that shot-noise limited estimations are implementable in constant time, e.g. by measuring $N$ single-particle observables simultaneously and neglecting the post-processing time overhead.

The time overhead of the phase estimation algorithm, as discussed in section 4.1.3, is dominated by the time of the controlled unitaries $\hat{C}_U(j)$ (reducible to the time-evolution $\hat{U}_t = e^{-i\hat{H}_h t}$ [70–74]), namely $t = 2^j \Delta t$ with $j = 0, 1, \ldots, \tilde{d} - 1$. The time required to implement the controlled unitaries for measuring each of the two phases, $\phi_c^{(N)}$ and $\phi_c^{(N+1)}$, is $\sum_{j=0}^{\tilde{d}-1} 2^j \Delta t = (2^{\tilde{d}} - 1)\Delta t = \mathcal{O}(2^d \Delta t)$, and therefore the running time for measuring both phases scales as $\Theta = 2^{d+1}\Delta t$. If standard estimations with shot-noise accuracy $\mathcal{O}(1/N)$ are repeated for time $\Theta$, the accuracy of the average estimation scales as $1/(\Theta N)$. Figure 6 shows that the accuracy (9) (red, transverse plane) is smaller than $1/(\Theta N)$ (blue surface) for small numbers of ancillary qubits: e.g., $d \geqslant 14$ for $N = 100$ and $\Delta t = \pi$, and in general $d \geqslant \log_2(32N\pi^2/\Delta t)$.

Compare now the magnetometric performances of protocol 2 with accuracies $\sim 1/N^2$, that can be achieved exploiting entangled states [91–93], continuous measurements [94–96], parametric down-conversion [97], compressive sensing [98], interactions with critical environments [6, 99, 100], phase estimation algorithms with adaptive techniques [101] or combined with SQUIDs [102], and sequential measurements [101, 103, 104]. For an easier comparison, assume that also these estimations are implemented in constant time, although the possibility to use powerful resources might require some time overhead, e.g., during the preparation of entangled states, the time needed for sequential measurements or for other preparation and post-processing operations.

Figure 6 compares the accuracy (9) also with the accuracy $1/N^2$ resulting from of a single estimation (green, upper surface) and with the accuracy of repetitions of such estimations for time $\Theta$ that is $1/(\Theta N^2)$ (purple, lower surface). Also these accuracies are outperformed by (9) for small numbers of ancillary qubits: $d \geqslant \log_2(4N\pi/\Delta t)$ for the accuracy $1/N^2$ and $d \geqslant \log_2(32N^2\pi^2/\Delta t)$ for the accuracy $1/(\Theta N^2)$, e.g., respectively $d \geqslant 9$ and $d \geqslant 20$ with $N = 100$ and $\Delta t = \pi$. Counting also the exact number of single- and two-particle operations of the phase estimation algorithm in the running time $\Theta$ does not produce perceivable changes in figure 6.

Protocol 2 can also reach exponentially small accuracy in $N$ and exponentially smaller than the other accuracies in figure 6, at the cost of polynomially many ancillary qubits $d = \mathcal{O}(\text{poly}(N))$ and exponential time $\Theta = \mathcal{O}(2^{\text{poly}(N)})$.

## 4.3 Probabilistic performances

The measurements of $H_h$ in the above protocols result in probabilistic probe preparations. We have discussed above that the eigenenergies and the corresponding eigenstates are sampled by repeated probe preparations, and the critical state is chosen exploiting the Hamiltonian spectral properties. If, however, a single energy measurement is performed, the probe state

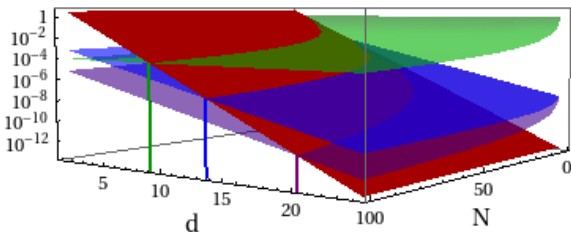

Figure 6: Accuracy (9) of protocol 2 (red, transverse plane), accuracy of shot-noise estimations repeated for time $\Theta = 2^{d+1}\Delta t$ (blue, middel surface), accuracy $1/N^2$ (green, upper surface), estimations with accuracy $1/N^2$ repeated for time $\Theta$ (purple, lower surface). We set $\Delta t = \pi$. Note the logarithmic scale of the vertical axes.

results in a random eigenstate, say the $k$-th eigenstate with eigenenergy $E_k$ and probability $p(E_k)$ given by the initial state. As a consequence, the metrological performances are also probabilistic. Therefore, the framework to be applied is that of probabilistic metrology [105], where the state after the probe preparation is the mixture

$$\sigma = \sum_k p(E_k)|E_k\rangle\langle E_k| \otimes |D_k\rangle\langle D_k|, \tag{11}$$

with the system energy eigenstates $|E_k\rangle$ and orthogonal detector states $|D_k\rangle$. In the phase estimation algorithm the ancillary qubits play the role of the detector because they store the energy values $E_k$ and their measurement provide an estimation of $E_k$.

The QFI of the state (11) is the average $\sum_k p(E_k)\mathcal{F}_h(E_k)$ [105], and is dominated by the energy eigenstates close to the critical one within an energy width $N\Sigma_E^*(h)$, whose QFI, $\mathcal{F}_h(E_k)$, scales as $N^\gamma$ (see figure 4). Remind that we have assumed $\xi = \gamma$ in agreement with our numerical results and with the characterisation of critical points. Therefore, we estimate the average QFI as

$$\overline{\mathcal{F}_h} \sim \sum_{\substack{k \text{ such that} \\ |E_k - E_c| \leqslant N\Sigma_E^*(h)}} p(E_k)\mathcal{F}_h(E_k) \sim \int_{E_c - \frac{1}{2}N\Sigma_E^*(h)}^{E_c + \frac{1}{2}N\Sigma_E^*(h)} dE\, \rho(E)p(E)\mathcal{F}_h(E). \tag{12}$$

In order to visualise the overlap of the QFI $\mathcal{F}_h(E_k)$ with $\rho(E)p(E)$,these functions are plotted in figure 7, with two choices for $p(E_k)$. The first case is $p(E_k) = e^{-\beta E_k}/\sum_k e^{-\beta E_k}$ and corresponds to the state before the phase estimation algorithm being the Gibbs state at large temperatures. The second case, namely $p(E_k) = \left|\langle E_k|(|\downarrow\rangle)^{\otimes N}\right|^2$, occurs when the state before the phase estimation algorithm has all spins down in the $z$ direction. Note that some functions are rescaled by suitable factors (see the legends and the caption) in order to plot all of them in the same figure. The peak of $\rho(E)p(E)$ decreases with increasing $N$, but the peak of $\mathcal{F}_h(E_k)$ increases with $N$ so that $\rho(E)p(E)\mathcal{F}_h(E_k)$ remains highly peaked around the critical energy. When the initial state is $|\downarrow\rangle^{\otimes N}$, for instance, the probability to obtain the critical eigenstate decays very slowly with $N$, i.e. $p(E_k) = \left|\langle E_k|(|\downarrow\rangle)^{\otimes N}\right|^2 = \mathcal{O}(N^{-0.06})$ [52]. Therefore the averaged QFI scales as $\overline{\mathcal{F}_h} \sim N^{\gamma - 0.06} \sim N^{2.01}$.

Assume now that the probability of each eigenstate contributing to the equation (12) scales as $p(E) \sim 1/N^\upsilon$ with $\upsilon > 0$. Using the scaling $\Sigma_E^*(h) \sim N^\mu$ and the expression (2), the probability $\pi$ to obtain an eigenstate with energy close to $E_c$ within a width $N\Sigma_E^*(h)$ is

$$\pi \sim \int_{E_c - \frac{1}{2}N\Sigma_E^*(h)}^{E_c + \frac{1}{2}N\Sigma_E^*(h)} dE\, \rho(E)p(E) \sim N^{1+\mu-\upsilon}\ln N, \tag{13}$$

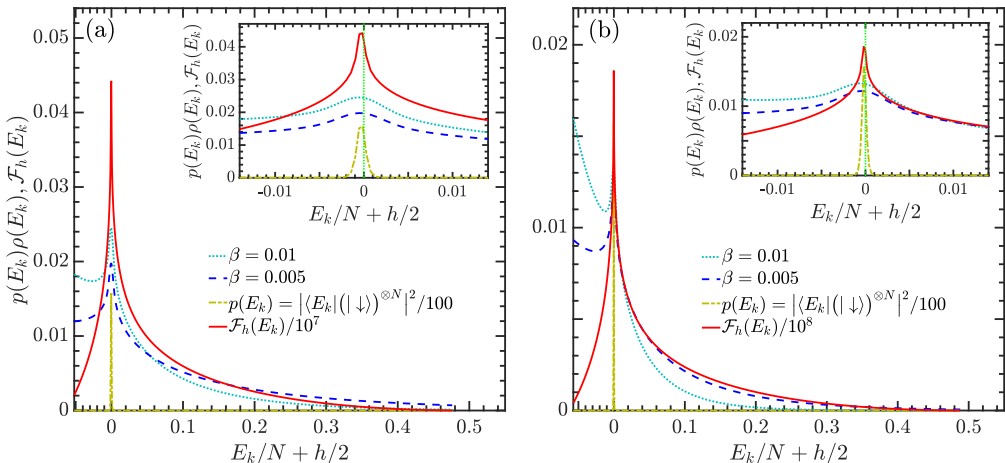

Figure 7: Comparison between the QFI $\mathcal{F}_h(E_k)$ and $p(E_k)\rho(E_k)$ as functions of $E_k/N + h/2$ with $h = 0.5$, $N = 800$ (panel (a)) and $N = 1600$ (panel (b)). Note that the QFI (solid, red line) is rescaled by $10^{-7}$ for $N = 800$ (a) and by $10^{-8}$ for $N = 1600$. The other curves are $p(E_k)\rho(E_k)$ with $p(E_k) = e^{-\beta E_k}/\sum_k e^{-\beta E_k}$ for $\beta = 0.01$ (dotted, cyan line) and $\beta = 0.005$ (dashed, blue line), and with $p(E_k) = \left|\langle E_k|(|\downarrow\rangle)^{\otimes N}\right|^2$ rescaled by $1/100$ (dot-dashed, yellow line). The insets are zooms of the peaks of the functions, and the green vertical lines therein correspond to the critical energy.

and the size scaling of the average QFI, $\overline{\mathcal{F}_h} \sim N^{\gamma-\upsilon+\mu+1}\ln N$, is superextensive if $\upsilon < \gamma + \mu \simeq 1.84$. The case $\upsilon = 1$ corresponds to the system being in the Gibbs state at high temperature before the phase estimation algorithm: $p(E) \sim 1/N$ as we have considered the subspace $s = N/2$.

Moreover, after $M$ repetitions of the energy measurement, the probability to obtain an eigenstate close to the critical one is increased to $\pi_M = 1 - (1 - \pi)^M$. If $\pi = \mathcal{O}(N^0)$ and $M \gg 1$, then $\pi_M \approx 1$. If $\pi$ goes to zero with increasing $N$, as in the above equation (13), and $M = a/\pi$ then $\pi_M \approx 1 - e^{-a}$.

We further recall that the QFI in figure 2 has a local minimum at $h = 0$ whose size scaling is fitted by $CN^\delta$, with $\delta \simeq 2.01$ irrespective of the energy eigenstates (see figure 3(b)). Therefore, given an unknown field $h$, any excited state $|E_k\rangle$, corresponding to the critical field $h_c^k$, can be used as probe for precision magnetometry if $h \in [-h_c^k, h_c^k]$. Within this interval, the QFI is lower bounded by $\mathcal{F}_{h=0}$, and the probabilistic energy measurement provide a probe with lower QFI only when energies above the critical one $E_c$ are measured. Therefore, the average QFI has the same size scaling of $\mathcal{F}_{h=0}$ with a constant prefactor that is the fraction of eigenstates with energy below the critical one $E_c$ [57]. Moreover, almost any Hamiltonian measurement outcome in the probe preparations provides a good probe for precision magnetometry of small magnetic fields.

## 5 Robustness of the magnetometric protocol

Our magnetometry performance is robust against imperfect probe preparations. First of all, several excited states at energies and magnetic fields around the critical ones serve as probes for enhanced precision magnetometry. The reason is that the QFI widths, $\Sigma_h$ and $\Sigma_E^*$, scale with $N$ much more slowly than the spacing between the critical fields of two adjacent excited states $D_k$, as shown in figures 3 and 4. Therefore, errors in the selection of the critical eigenstate

does not substantially reduce the metrological performances. In the following, we show the robustness of the magnetometric performances under different sources of noise.

## 5.1 Incoherent noise

We now investigate the effect of noise in the algorithm or in the detectors, such that the detector states $|D_k\rangle$ in equation (11) are the same for different $k$. In order to simplify the notation, assume the worst case where the states $|D_k\rangle$ are the same for all $k$. The state after the probe preparation is then $\sigma = \sum_k p(E_k)|E_k\rangle\langle E_k|$ and its QFI is

$$
\begin{aligned}
\mathcal{F}_h(\sigma) &= \sum_k \frac{\left(\partial_h p(E_k)\right)^2}{p(E_k)} + 2\sum_{\substack{k,l \\ k\neq l}} \frac{\left(p(E_l)-p(E_k)\right)^2}{p(E_k)+p(E_l)}\left|\langle E_k|\left(\partial_h|E_l\rangle\right)\right|^2 \\
&\simeq \int dE\rho(E)\frac{\left(\partial_h p(E)\right)^2}{p(E)} + 2\fint dE\rho(E)dE'\rho(E')\frac{\left(p(E)-p(E')\right)^2}{p(E)+p(E)}\left|\langle E'|\left(\partial_h|E\rangle\right)\right|^2 . \quad (14)
\end{aligned}
$$

Consider that the state $\sigma$ is well-localised around the excited state $|E_{\widetilde{k}}\rangle$ with energy closest to $E_c$, i.e. $E_{\widetilde{k}} \xrightarrow[N\gg 1]{} E_c$ and $p(E_{\widetilde{k}}) \gg p(E_{k\neq\widetilde{k}}) = \mathcal{O}(1/N^\alpha)$ with $\alpha > 0$. Expanding the second sum of equation (14) for large $N$, the first order term comes from the contributions with either $k = \widetilde{k}$ or $l = \widetilde{k}$ and is proportional to the QFI of the pure state $|E_{\widetilde{k}}\rangle$ (compare this term to equation (A.4) in appendix A). We then obtain superextensive QFI:

$$
\mathcal{F}_h(\sigma) = \sum_k \frac{\left(\partial_h p(E_k)\right)^2}{p(E_k)} + \mathcal{F}_h(E_{\widetilde{k}})\left(p(E_{\widetilde{k}}) + \mathcal{O}(N^{-\alpha})\right) \sim p(E_{\widetilde{k}})N^\gamma . \quad (15)
$$

This case encompasses our protocols with all spins pointing down in the $z$ direction at the beginning of the probe preparation step, where $p(E_{\widetilde{k}}) = \mathcal{O}(N^{-0.06})$ [52].

The QFI remains superextensive not only for an incoherent perturbation of the desired excited state, but also for moderate noise. Indeed, consider probabilities

$$
p(E_k) = \begin{cases} \mathcal{O}\left(\dfrac{1}{N^v}\right) & \text{if } |E_k - E_c| \lesssim \Delta = \mathcal{O}(N^{v'}) \\ \mathcal{O}\left(\dfrac{1}{N^\alpha}\right) & \text{if } |E_k - E_c| \gtrsim \Delta = \mathcal{O}(N^{v'}) \end{cases} , \quad (16)
$$

with $\alpha > v > 0$ so that $1/N^v \gg 1/N^\alpha$ for large $N$. The normalisation

$$
\begin{aligned}
1 &= \sum_k p(E_k) \simeq \int dE\rho(E)p(E) = \int_{|E-E_c|\lesssim\Delta} dE\rho(E)\underbrace{p(E)}_{\mathcal{O}(\frac{1}{N^v})} + \int_{|E-E_c|\gtrsim\Delta} dE\rho(E)\underbrace{p(E)}_{\mathcal{O}(\frac{1}{N^\alpha})} \\
&= \mathcal{O}(N^{v'-v}) + \mathcal{O}(N^{1-v'-\alpha}) \quad (17)
\end{aligned}
$$

imposes that $v' - v \leqslant 0$ and $1 - v' - \alpha \leqslant 0$. For the sake of simplify, we neglect the logarithmic factors due to the behaviour of $\rho(E)$ around $E_c$, which can be reabsorbed as logarithmic corrections of the above and the following scalings. Moreover,

$$
\min_{\substack{k:|E_k-E_c|\lesssim\Delta \\ l:\Delta\lesssim|E_l-E_c|\lesssim\frac{1}{2}N\Sigma_E^*(h)}} \left(p(E_k)-p(E_l)\right) = \mathcal{O}\left(\frac{1}{N^v}\right), \quad (18)
$$

and we also assume

$$\min_{\substack{k,l:\\|E_{k,l}-E_c|\lesssim\Delta}} \left(p(E_k)-p(E_l)\right) = \mathcal{O}\left(\frac{1}{N^\iota}\right). \tag{19}$$

The following consistency relation

$$\max_{k:|E_k-E_c|\lesssim\Delta} p(E_k) - \min_{l:|E_l-E_c|\lesssim\Delta} p(E_l) = \mathcal{O}\left(\frac{1}{N^\upsilon}\right)$$

$$\gtrsim \min_{\substack{k,l:\\|E_{k,l}-E_c|\lesssim\Delta}} \left(p(E_k)-p(E_l)\right) \int_{E_c-\Delta}^{E_c+\Delta} dE\rho(E) = \mathcal{O}\left(N^{\upsilon'-\iota}\right) \tag{20}$$

implies $\iota \geqslant \upsilon + \upsilon'$.

Consider also $\upsilon' < 1 + \mu$, so that $\Delta < N\Sigma_E^*/2$ and we can apply the scaling of the energy differences around $E_c$ discussed at the end of section 3: for $|E_{k,l} - E_c| \lesssim \Delta$,

$$\left|\langle E_k|\left(\partial_h|E_l\rangle\right)\right|^2 = \frac{\left|\langle E_k|\hat{S}_z|E_l\rangle\right|^2}{(E_k-E_l)^2} = \mathcal{O}\left(\frac{N^\nu}{\ln N}\right), \tag{21}$$

where we have used equation (A.5) in appendix A, and recall $\nu = \gamma - \mu - 1$. Using the above scalings, we estimate

$$\mathcal{F}_h(\sigma) \simeq \int dE\rho(E)\frac{(\partial_h p(E))^2}{p(E)} + 2 \fint_{\substack{|E-E_c|\lesssim\Delta\\|E'-E_c|\lesssim\Delta}} dE\rho(E)dE'\rho(E')\frac{(p(E)-p(E'))^2}{p(E)+p(E')}\left|\langle E'|\left(\partial_h|E\rangle\right)\right|^2$$

$$+ 4 \fint_{\substack{\Delta\lesssim|E-E_c|\lesssim N\Sigma_E^*(h)\\|E'-E_c|\lesssim\Delta}} dE\rho(E)dE'\rho(E')\frac{(p(E)-p(E'))^2}{p(E)+p(E')}\left|\langle E'|\left(\partial_h|E\rangle\right)\right|^2$$

$$\geqslant \mathcal{O}\left(N^{\nu+\upsilon+2\upsilon'-2\iota}\right) + \mathcal{O}\left(N^{\nu-\upsilon+\upsilon'+\mu+1}\right), \tag{22}$$

where we have used the symmetry under the exchange $E \leftrightarrow E'$ in the third integral. The first and the second terms after the last equality in equation (22) are the estimates of the second and the third integrals respectively, while the first integral scales as $\mathcal{O}(N^0)$ if $\partial_h p(E) \sim p(E)$. In the first equality of equation (22) we have neglected contributions from energies $E$ such that $|E - E_c| \gtrsim N\Sigma_E^*(h)$. Indeed, they are subleading orders because $\left|\langle E'|\left(\partial_h|E\rangle\right)\right|^2 = \mathcal{O}(N^0)$, as follows from equation (A.5) in appendix A together with the estimates $\langle E_k|\hat{S}_z|E_l\rangle = \mathcal{O}(N)$ and $E_k - E_l = \mathcal{O}(N)$ for these energies.

The constraints $\iota \geqslant \upsilon + \upsilon'$ and $\upsilon' < 1 + \mu$ imply that the second estimate in equation (22) dominates: $\nu + \upsilon + 2\upsilon' - 2\iota < \nu - \upsilon + \upsilon' + \mu + 1 = \gamma - \upsilon + \upsilon'$. Therefore, the QFI of the noisy state $\sigma$ is superextensive if $\gamma - \upsilon + \upsilon' > 1$. This condition is compatible with the above constraints, including $\upsilon' - \upsilon \leqslant 0$, and the maximum exponent, i.e. $\gamma$, is reached for $\upsilon = \upsilon'$. Figure 8 shows the exponent $\gamma - \upsilon + \upsilon'$ in the region defined by the constraints. Interestingly, also the exponent $\nu + \upsilon + 2\upsilon' - 2\iota$ can be larger than one, compatibly with the constraints, and its maximum is $\nu \simeq 1.3$ when $\upsilon = \upsilon' = \iota = 0$. Note that if we relax the assumption $\upsilon' < 1 + \mu$, we only have the first two integrals in the estimate (22) with $\Delta$ replaced by $N\Sigma_E^*(h)$. The size scaling of the QFI is then $\mathcal{O}\left(N^{\nu+\upsilon+2\mu+2-2\iota}\right)$ which cannot be superextensive given the remaining constraints.

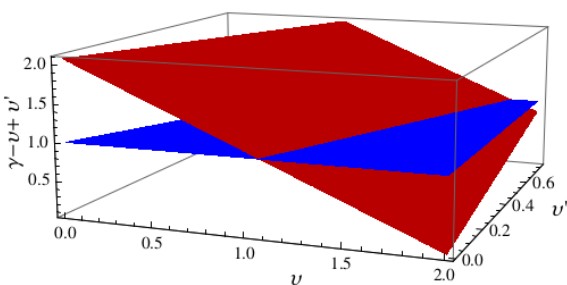

Figure 8: Exponent $\gamma - \upsilon + \upsilon'$ (red, transverse plane) in the region defined by $\upsilon' - \upsilon \leqslant 0$ and $\upsilon' < 1 + \mu$, compared with the value 1 (blue, horizontal plane).

## 5.2 Coherent noise

Here, we consider noisy probe preparations realised with a pure state $|\phi\rangle = \sum_k c_k |E_k\rangle$. An example is the state after the phase estimation algorithm with a finite accuracy. The QFI is

$$
\mathcal{F}_h(\phi) = 4 \sum_k \left| \partial_h c_k \right|^2 + 4 \sum_{k,l} \overline{c}_k c_l \big( \partial_h \langle E_k| \big)\big( \partial_h |E_l\rangle \big) + 4 \sum_{\substack{k,l \\ k \neq l}} \Big( \big( \partial_h \overline{c_k} \big) c_l \langle E_k| \big( \partial_h |E_l\rangle \big) + \text{c.c.} \Big)
$$
$$
+ 4 \left( \sum_k \overline{c_k}\, \partial_h c_k + \sum_{\substack{k,l \\ k \neq l}} \overline{c_k}\, c(E_l) \langle E_k| \big( \partial_h |E_l\rangle \big) \right)^2 .
\tag{23}
$$

Similarly to the case of incoherent noise, consider a state $|\phi\rangle$ well-localised around $|E_{\tilde{k}}\rangle$ such that $E_{\tilde{k}} \xrightarrow[N \gg 1]{} E_c$ and $|c_{\tilde{k}}|^2 \gg |c_{k \neq \tilde{k}}|^2 = \mathcal{O}(1/N^\alpha)$ with $\alpha > 0$. Expanding for large $N$, the QFI is approximated by the following superextensive scaling

$$
\mathcal{F}_h(\phi) = 4 \sum_k \left| \partial_h c_k \right|^2 + 4 \left( \sum_k \overline{c}_k \partial_h c_k \right)^2 + \mathcal{F}_h(E_{\tilde{k}}) \left( |c_{\tilde{k}}|^2 + \mathcal{O}\left( N^{-\frac{\alpha}{2}} \right) \right) \sim |c_{\tilde{k}}|^2 N^\gamma .
\tag{24}
$$

For instance, if the system before the probe preparation step is in the state with all spins pointing down in the $z$ direction, the overlap with the critical eigenstate is $|c_{\tilde{k}}|^2 = \mathcal{O}(N^{-0.06})$ [52].

## 5.3 Hamiltonian perturbations

Consider now imperfections in the system Hamiltonian that, due to the high precision of experimental realisations [39–46], can be modelled by perturbations of the Hamiltonian $\hat{H}'_h = \hat{H}_h + g \hat{H}_{\text{pert}}$. Examples of perturbations are transverse fields or interactions, including deviations from the infinite range interactions in (1), $\sum_{j,l} \big( |i-j|^{-\alpha} - 1 \big) \hat{\sigma}_x^i \sigma_x^j / N$.

Applying perturbation theory, the perturbed eigenvalues are

$$
E'_{k'} = E_k + \sum_{j \geqslant 1} g^j E_k^{(j)}, \qquad E_k^{(1)} = \langle E_k | \hat{H}_{\text{pert}} | E_k \rangle .
\tag{25}
$$

The perturbed derivatives in equation (10) are

$$
\partial_h E'_k = \partial_h E_k + 2 g \, \text{Re} \langle E_k | \hat{H}_{\text{pert}} \, \partial_h | E_k \rangle + \mathcal{O}(g^2) = \partial_h E_k + 2 g \, \langle E_k | \hat{A} \hat{S}_z | E_k \rangle + \mathcal{O}(g^2),
\tag{26}
$$

where we have used equation (A.5) and

$$\hat{A} = \hat{H}_{\text{pert}} \sum_{n \neq k} \frac{|E_n\rangle\langle E_n|}{E_k - E_n} \, . \tag{27}$$

The perturbed magnetometric variance in protocol 2 at finite size is $\epsilon^2 / \left(\partial_h \Delta \phi_k'\right)^2$ with

$$\frac{\partial_h \Delta \phi_k'}{\partial_h \Delta \phi_k} = 1 + \frac{2 g \Delta t}{\partial_h \Delta \phi_k} \left( \text{Re} \left\langle E_k^{(N)} \big| \hat{H}_{\text{pert}} \, \partial_h \big| E_k^{(N)} \right\rangle - \text{Re} \left\langle E_k^{(N+1)} \big| \hat{H}_{\text{pert}} \, \partial_h \big| E_k^{(N+1)} \right\rangle \right) + \mathcal{O}(g^2) \, . \tag{28}$$

Applying the Cauchy-Schwatz inequality, we bound the first order contribution:

$$\left| \langle E_k | \hat{H}_{\text{pert}} \, \partial_h | E_k \rangle \right| \leqslant \sqrt{\langle E_k | \hat{A}^\dagger \hat{A} | E_k \rangle \langle E_k | \hat{S}_z^2 | E_k \rangle} = \mathcal{O}(N) \sqrt{\lim_{g \to 0} \mathcal{F}_g(E_k')} \, , \tag{29}$$

where we have used $\langle S_z^2 \rangle \leqslant N^2$, and $\mathcal{F}_g(E_k')$ is the expression (3) for the QFI of the eigenstates of the perturbed Hamiltonian $H_h'$ with the derivative $\partial_h$ replaced by $\partial_g$.

Remind that $\partial_h E_k^{(N)} = C N^\kappa$ implies $\partial_h \Delta \phi_k = \mathcal{O}(N^0)$, as discussed in section 4.2. Assuming that the QFI with respect to the perturbative parameter $g$ is at most extensive, i.e. $\lim_{g \to 0} \mathcal{F}_g(E_k') = \mathcal{O}(N)$ in equation (29), the first order of the expansion (28) is bounded from above by $\mathcal{O}\big(\sqrt{N}\big)$, and thus is a small correction if $g = \tilde{g}/\sqrt{N}$ with $\tilde{g} \ll 1$. This scaling of $g$, with $N$-independent $\tilde{g}$, is detectable using classical metrology that achieves shot-noise accuracy. Finally, the assumption $\lim_{g \to 0} \mathcal{F}_g(E_k') = \mathcal{O}(N)$ holds if there is not a ESQPT at $g = 0$ for the Hamiltonian $H_h'$ and its eigenstate $|E_k'\rangle$. Therefore, in order to completely spoil the magnetometric performances, the perturbed Hamiltonian $H_h'$ should exhibit singularities at $g = 0$ for all its eigenstates. This condition is not met by a typical perturbation $H_{\text{pert}}$.

# 6 Conclusions

In conclusion, we have characterised the excited state quantum phase transition of the Lipkin-Meshkov-Glick model by means of the Fisher information. The ESQPT occurs in the interval of the magnetic field $h \in [-h_c, h_c]$: each excited state has a different critical field. The QFI has broad peaks whose maximum values $\mathcal{F}_{h,m} \sim N^{2.07}$ are located at critical fields and energy.

The spectral properties induced by the ESQPT are responsible for the QFI peaks, and also allows us to design two efficient schemes for precision magnetometry, which exploit a small quantum register performing the quantum phase estimation algorithm. In many quantum metrological schemes, entanglement is the key resource, and our probe states, i.e. Hamiltonian eigenstates, are indeed entangled. Nevertheless, our results lead us to suggest that the critical behaviour of Hamiltonian eigenstates is the key resource for enhanced precision magnetometry discussed in this paper. The first protocol achieves sub-shot-noise accuracy $\sim 1/N^{1.5}$ with register size and running time constant in $N$. The second protocol can achieve any polynomially small (in $N$) accuracy with logarithmically many register qubits and polynomial running time. In particular, shot-noise estimations repeated for the whole duration of our protocol, and estimations based on entangled states are outperformed with small numbers of register qubits. We have also shown that the performances of magnetometric protocols based on the ESQPT are robust against several noise sources, thanks to the spectral properties and the broadness of the QFI peaks.

Similar behaviours are expected in extended models [106–108], and in other models with ESQPTs which share similar spectral properties [22, 28, 30, 32, 109, 110]. Our analysis can also be applied to estimate microscopic parameters of physical systems that can be mapped

to these models. There are several instances of these parameters: the kinetic energy, the interaction strength, or the critical temperature in the strong coupling limit of superconducting systems [33–35, 111, 112]; interaction strengths in nuclear systems [36–38]; tunneling or the interaction strength in Bosonic Josephson junctions [41, 113]; detunings, pump and driving laser magnitudes, or atom-photon interaction in a Bose-Einstein condensate in an optical cavity [47]; atom-pump detuning and interaction, pump driving, or photon decay rate in cavity QED [48]; phonon and Rabi frequencies, and laser detunings in trapped ions [46]. Eingenstates of the LMG model at small size can be simulated also with variational hybrid quantum-classical algorithms with low depth on superconducting transmon qubits, and signatures of their phase transitions can be computed [114]. The robustness of magnetometric protocols based on the ESQPT and the aforementioned variety of platforms for realizing or simulating the LMG model make these protocols good candidates for NISQ devices.

# Acknowledgements

U. M. acknowledges interesting and helpful discussions with Andrea Trombettoni.

**Funding information**    Q. W. acknowledges support from the Slovenian Research and Innovation Agency (ARIS) under the Grant Nos. J1-4387 and P1-0306, the National Science Foundation of China under grant No. 11805165, and Zhejiang Provincial Nature Science Foundation under Grant No. LY20A050001, U. M. acknowledges support from the European Union's Horizon 2020 research and innovation programme under the Marie Skłodowska-Curie grant agreement No. 754496 - FELLINI.

# A    Quantum Fisher information

We briefly derive some properties of the QFI [1–4]. Consider a density matrix $\rho(h)$ depending on the parameter $h$. Its change with respect to $h$ can be expressed as

$$\partial_h \hat{\rho}(h) = \frac{1}{2}\big(\hat{L}_h \hat{\rho}(h) + \hat{\rho}(h)\hat{L}_h\big), \tag{A.1}$$

where $\hat{L}_h$ is the symmetric logarithmic derivative. The QFI is

$$\mathcal{F}_h = \text{Tr}[\hat{\rho}(h)\hat{L}_h^2] = \text{Tr}[\hat{L}_h \partial_h \hat{\rho}(h)]. \tag{A.2}$$

From the spectral decomposition of the system state $\hat{\rho}(h) = \sum_k \lambda_k(h)|\psi_k(h)\rangle\langle\psi_k(h)|$, a more explicit formula for the QFI is

$$\mathcal{F}_h = 2 \sum_{\substack{k,l:\\ \lambda_k+\lambda_l>0}} \frac{|\langle\psi_k|\partial_h\hat{\rho}(h)|\psi_l\rangle|^2}{\lambda_k + \lambda_l}, \tag{A.3}$$

where the sum carries over only terms that satisfy $\lambda_k + \lambda_l \neq 0$. Equation (A.3) for pure states $\hat{\rho}(h) = |\psi(h)\rangle\langle\psi(h)|$ becomes

$$\mathcal{F}_h = 4\big(\partial_h\langle\psi|\big)\big(\partial_h|\psi\rangle\big) + 4\big(\langle\psi|\big(\partial_h|\psi\rangle\big)\big)^2. \tag{A.4}$$

When the pure state is a Hamiltonian eigenstate $|k\rangle$, we can use perturbation theory [115] where the perturbation is a small increment of the magnetic field term in the Hamiltonian (1),

to compute

$$\partial_h |E_k\rangle = \sum_{n \neq k} \frac{\langle E_n | \hat{S}_z | E_k \rangle}{E_k - E_n} |E_n\rangle, \tag{A.5}$$

which satisfy $\langle E_k | (\partial_h |E_k\rangle) = 0$, then obtaining the QFI in equation (3).

## B  Excited state quantum phase transitions

We provide here a few additional details on the phase transitions of the LMG model with the Hamiltonian (1).

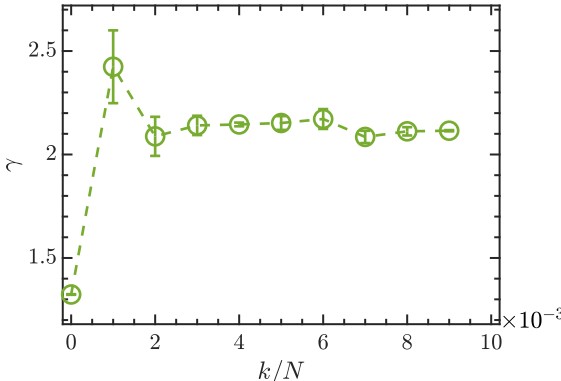

Figure 9: The value of the power law exponent $\gamma$ in $\mathcal{F}_{h,m} = CN^\gamma$ for the ten lowest Hamiltonian eigenstates in the eigenspace of $\hat{S}^2$ with $s = \frac{N}{2} = 500$. $k = 0$ denotes the ground state.

We first notice that the power law scaling of the QFI $\mathcal{F}_{h,m} = CN^{2.07}$ at critical fields (see figure 3(a)) does not hold for the quantum phase transition in the ground state. Indeed, we fitted the size scaling of the QFI at the critical field $h_c = \pm 1$ for the ground state phase transition, $\mathcal{F}_{h,m}^G \sim N^{1.33}$, that is consistent with the results discussed in [116]. Note that the exponent of $\mathcal{F}_{h,m}^G$ is smaller than the exponent of the QFI in ESQPTs. In figure 9, we plot the exponent of the power law $\mathcal{F}_{h,m} = CN^\gamma$ for the ten lowest Hamiltonian eigenstates in the eigenspace of $\hat{S}^2$ with $s = \frac{N}{2}$.

For completeness, we plot in figure 10 the multiplicative constants $C$ of all power law fits shown in Figs. 3 and 4.

We now characterize more in detail the size scaling of the QFI around its peaks. Figure 11 shows the QFI rescaled by $N^2$, namely $\mathcal{F}_h/N^2$, for different $N$ as a function of the magnetic field and the eigenenergy. Notice that the curves at different $N$ overlap except around the peak. This behaviour is compatible with the fits of the peaks plotted in figures 3 and 4, $\mathcal{F}_{h,m} = \mathcal{O}(N^\gamma)$ and $\mathcal{F}_{h,m}^* = \mathcal{O}(N^\xi)$, which show exponents $\gamma \simeq \xi \simeq 2.07$ slightly larger than 2. In order to emphasize the numerical evidence of such scaling, we show in figure 12 that the peak values $\mathcal{F}_{h,m}/N^2$ and $\mathcal{F}_{h,m}^*/N^2$ slowly increase with $N$ for several eigenstates and magnetic fields.

## C  Preparation of Gibbs states

We now provide some details on the preparation of the Gibbs state used in the probe preparation of protocol 1. The Gibbs state can be achieved through the interaction with a thermal bath.

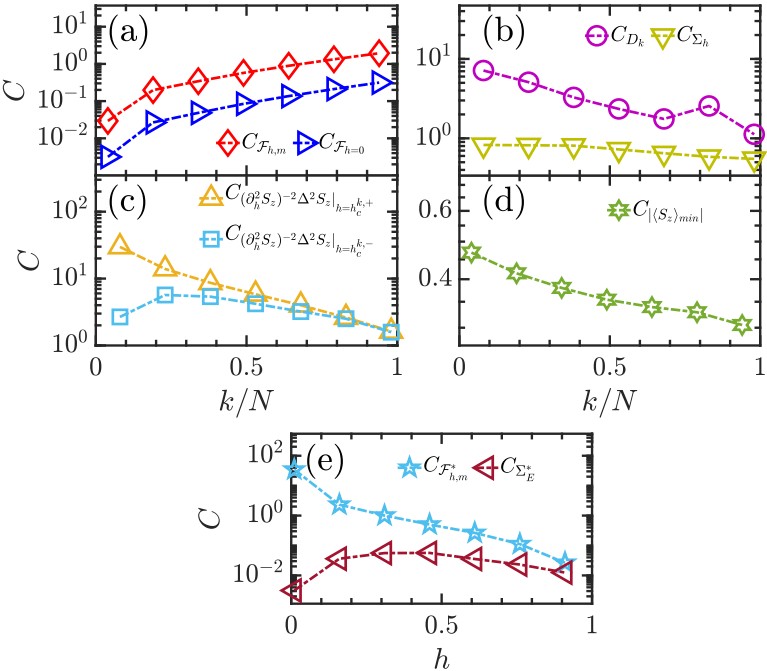

Figure 10: The values of the constant $C$ of different power law fits for several Hamiltonian eigenstates (panels (a)-(d)) and several control parameters (panel (e)).

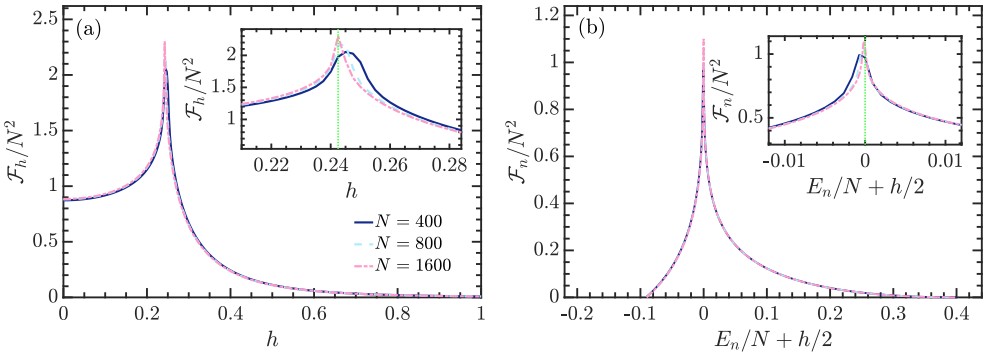

Figure 11: Rescaled QFI $\mathcal{F}_h(E_n)/N^2$ for $N = 400$ (blue, continuous curve), $N = 800$ (cyan, dashed curve), and $N = 1600$ (pink, dot-dashed curve). Panel (a): $\mathcal{F}_h(E_n)/N^2$ as a function of $h$ with $n = 0.2N$. Panel (b): $\mathcal{F}_h(E_n)/N^2$ as a function of $E_n$ with $h = 0.4$. The insets are zooms of the peaks, and the green vertical lines therein correspond to the critical field in panel (a) and to the critical energy in panel (b).

The reduced dynamics of the system is derived within Davies' weak coupling approach [64,65], consistently assuming

- that the system-bath interaction is of the general form $\hat{H}_I = \lambda \sum_j \hat{A}_j \otimes \hat{B}_j$ where $\hat{A}_j$ are operators of the system, $\hat{B}_j$ are operators of the bath, and with small $\lambda \ll 1$;

- that, given the bath Hamiltonian $\hat{H}_B$, the bath is initially in the thermal state $\hat{\rho}_B = \frac{e^{-\beta \hat{H}_B}}{\text{Tr} e^{-\beta \hat{H}_B}}$ uncorrelated with the system, and that the bath is so large that it cannot be substantially perturbed by the interaction with the system;

- that the bath dynamics is so fast compared to the system time scale to apply the Markovian approximation.



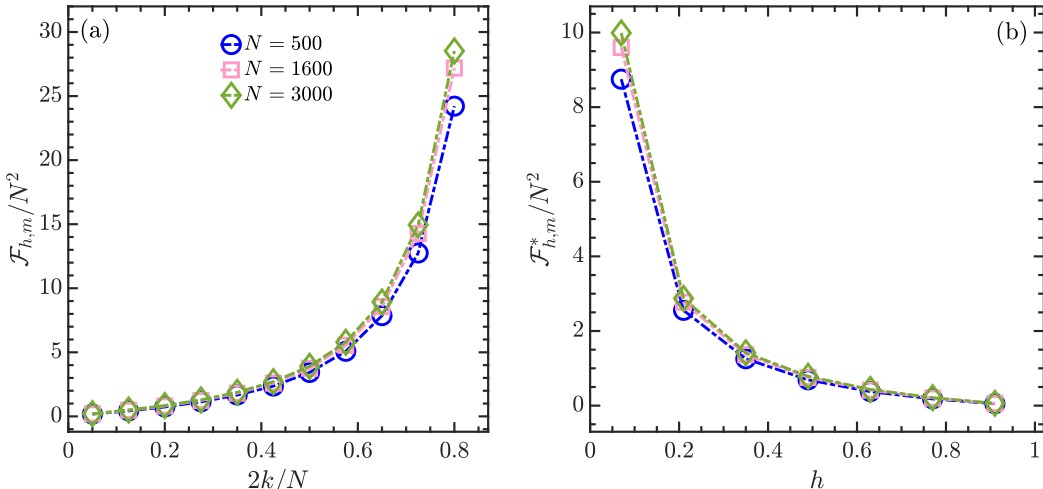

Figure 12: Panel (a): the peak value in figure 11(a) $\mathcal{F}_{h,m}(E_k)$ as a function of $2k/N$. Panel (b): the peak value in figure 11(b) $\mathcal{F}^*_{h,m}(E_k)$ as a function of $h$. The peak values are plotted for $N = 500$ (blue circles), $N = 1600$ (pink squares), and $N = 3000$ (green diamonds).

This approach results in the following generalisation of the Schrödinger equation, generically called master equation,

$$
\frac{\mathrm{d}}{\mathrm{d}t}\hat{\rho} = -i\left[\hat{H}_h + \lambda^2 \sum_{j,l,\omega} s_{j,l}\hat{A}_j(\omega)\hat{A}^\dagger_l(\omega), \hat{\rho}\right]
$$
$$
+ \lambda^2 \sum_{j,l,\omega} h_{j,l}\left(\hat{A}^\dagger_l(\omega)\hat{\rho}\hat{A}_j(\omega) - \frac{1}{2}\{\hat{A}_j(\omega)\hat{A}^\dagger_l(\omega), \hat{\rho}\}\right), \tag{C.1}
$$

where, given the eigenvalues $\{\epsilon_a\}_a$ of the system Hamiltoninan $\hat{H}_h$ and the corresponding eigenprojectors $\{\hat{\Pi}_a\}_a$,

$$
\hat{A}_j(\omega) = \sum_{\epsilon_a - \epsilon_b = \omega} \hat{\Pi}_a \hat{A}_j \hat{\Pi}_b, \tag{C.2}
$$

and $h_{j,l}(\omega)$ and $s_{j,l}(\omega)$ can be obtained from, respectively, the real and the imaginary part of the one-sided Fourier transform of two-point correlation functions of bath operators $\hat{B}_j$:

$$
\Gamma_{j,l}(\omega) = \frac{h_{j,l}(\omega)}{2} + i s_{j,l}(\omega) = \int_0^\infty dt\, e^{i\omega t}\mathrm{Tr}\left(\hat{\rho}_B\, e^{it\hat{H}_B}\hat{B}_j e^{-it\hat{H}_B}\hat{B}_l\right). \tag{C.3}
$$

Diagonalising the positive matrix $[h_{j,l}(\omega)]_{j,l}$, the second line of (C.1) can be rewritten in the Gorini-Kossakowski-Sudarshan-Lindblad form

$$
\sum_j \gamma_j\left(\hat{L}_j\hat{\rho}\hat{L}^\dagger_j - \frac{1}{2}\{\hat{L}^\dagger_j\hat{L}_j, \hat{\rho}\}\right), \tag{C.4}
$$

that guarantees the complete positivity of the time-evolution [65].

The time-evolution that solves the master equation (C.1) relaxes to the Gibbs state of the system Hamiltonian $\hat{H}_h$, if it is the unique stationary state. A sufficient condition is that operators that commute with all $\hat{A}_j(\omega)$ and $\hat{A}^\dagger_j(\omega)$ are proportional to the identity. The system

thermalizes in the subspace with $s = N/2$ if the initial state belongs to this subspace and if all the operators $\hat{A}_j$ commute with $\hat{S}^2$.

An alternative preparation procedure without the condition that $\hat{A}_j$ commute with $\hat{S}^2$, consists of a thermalizing master equation with the new Hamiltonian $\hat{H}_h - \chi \hat{S}^2$. The derivation of the master equation (C.1) is invariant under the shift $\hat{H}_h \to \hat{H}_h - \chi \hat{S}^2$ because $\hat{S}^2$ commutes with $\hat{H}$ and shares eigenprojectors with $\hat{H}_h$. If $\chi \gg 1$ all eigenspaces of $\hat{S}^2$ with $s \neq N/2$ are exponentially suppressed. Instead of $-\chi \hat{S}^2$, any additional Hamiltonian term $-\chi f(\hat{S}^2)$, with $f(\cdot)$ a monotonically increasing function and $\chi > 0$, plays the same role.

# D   Phase estimation algorithm

The phase estimation algorithm measures the phases of $\hat{U}_{\Delta t}$, that are $\phi_k = E_k \Delta t \mod 2\pi$ with accuracy $\epsilon = \pi/2^d$. If the condition $\Delta t < 2\pi/\max_{k,k'}\{|E_k - E_{k'}|\} = \mathcal{O}(1/N)$ doesn't hold, the distibution of phases $\phi_k$ is different from that of eigenenergies $E_k$. In particular, different eigenenergies, say $E_k$ and $E_l$ corresponding to phases $\phi_k$ and $\phi_l$, can be confused if

$$|\phi_k - \phi_l| = \left|(E_k - E_l)\Delta t - 2\pi m\right| \leqslant \epsilon \tag{D.1}$$

with $m \in \mathbf{Z}$, or equivalently

$$\left|\frac{E_k - E_l}{2\pi m}\Delta t - 1\right| \leqslant \frac{\epsilon}{2\pi|m|} = \frac{1}{2^{d+1}|m|}. \tag{D.2}$$

Note that, if $\Delta t = \mathcal{O}(N^0)$, then equation (D.2) implies that $(E_k - E_l)/m = \mathcal{O}(N^0)$: when at least one of $E_k$ and $E_l$ are away from the critical energy, then both $E_k - E_l$ and $m$ scale as $\mathcal{O}(N)$; if both $E_k$ and $E_l$ are close to the critical energy, then the average scaling of both $E_k - E_l$ and $m$ is $\mathcal{O}(N^{1-\nu/2}\sqrt{\ln N})$ with $1 - \nu/2 \simeq 0.35$ (see the discussion at the end of section 3. Moreover, $d$ can be chosen independently of $N$ in protocol 1 or grows polynomially in $N$ in protocol 2. Therefore, if $\Delta t$ is random in a constant range, say $[0,1]$, the eigenenergies $E_k$ and $E_l$ are confused for $\Delta t$ ranging in a finite number of intervals with amplitudes $2^{-d}\pi/(E_k - E_l)$ that approach zero for large $N$, therefore with vanishingly small probability.

As a consequence, consider two errors that can occur in our protocols due to the periodicity modulo $2\pi$ of the phase estimation algorithm. The first occurs when an eigenvalue $E_k$ is confused with those close to $E_c = -hN/2$. The second error occurs when one observes a spurious peak in the density of measured phases, beyond that corresponding to $E_c$. The above cosiderations imply that the probability of these errors vanishes for large $N$. Moreover, the spurious peak can be confused with that corresponding to $E_c$ if they have similar height, that shows a logarithmic divergence in the thermodynamic limit as in equation (2). Thus, it is even less probable that infinitely many energy eigenstates far from $E_c$ bunch together for large $N$.

# E   Hellmann-Feynman theorem

In this appendix, we derive some consequences of the Hellmann-Feynman theorem and of the degeneracy $E_k = \tilde{E}_k$ for $h \leqslant h_c^k$, or equivalently $E_k \leqslant E_c$. The Hellmann-Feynman theorem states the following

$$\begin{aligned} dE_k &= d\langle E_k|\hat{H}_h|E_k\rangle = \langle E_k|d\hat{H}_h|E_k\rangle + \left(d\langle E_k|\right)\hat{H}_h|E_k\rangle + \langle E_k|\hat{H}_h\left(d|E_k\rangle\right) \\ &= \langle E_k|d\hat{H}_h|E_k\rangle + E_k\left(\left(d\langle E_k|\right)|E_k\rangle + \langle E_k|\left(d|E_k\rangle\right)\right) = \langle E_k|d\hat{H}_h|E_k\rangle, \end{aligned} \tag{E.1}$$

where we have used $\hat{H}|E_k\rangle = E_k|E_k\rangle$, and $\langle E_k|\big(\mathrm{d}|E_k\rangle\big) = 0$ from perturbation theory (see also equation (A.5) in appendix A). The Hellmann-Feynman theorem allows us to write derivatives of Hamiltonian eigenvalues as expectation values, i.e.,

$$\partial_h E_k = \langle E_k|\partial_h \hat{H}_h|E_k\rangle = \langle E_k|\hat{S}_z|E_k\rangle. \tag{E.2}$$

Equations (E.1) and (E.2) remain valid with $E_k$ and $|E_k\rangle$ replaced by $\tilde{E}_k$ and $|\tilde{E}_k\rangle$ respectively. As the $k$-th even and the $k$-th odd eigenvalues equal each other for $h \leqslant h_c^k$, also their derivatives are equal in the same parameter range. Therefore, the Hellmann-Feynman theorem (see equation (E.2)) implies

$$\langle E_k|\hat{S}_z|E_k\rangle = \langle \tilde{E}_k|\hat{S}_z|\tilde{E}_k\rangle, \qquad \text{for} \quad h \leqslant h_c^k. \tag{E.3}$$

Equation (E.1) can be generalized to any function of the Hamiltonian $f(\hat{H}_h)$, following the same steps:

$$\mathrm{d}f(E_k) = \mathrm{d}\langle E_k|f(\hat{H}_h)|E_k\rangle = \langle E_k|\mathrm{d}f(\hat{H}_h)|E_k\rangle. \tag{E.4}$$

If $f(\hat{H}_h) = \hat{H}_h^2/h$, the equality (E.4) implies the following relation

$$\partial_h\left(\frac{E_k^2}{h}\right) = \langle E_k|\partial_h\left(\frac{\hat{H}_h^2}{h}\right)|E_k\rangle = \langle E_k|\hat{S}_z^2|E_k\rangle - \frac{1}{h^2 N^2}\langle E_k|\hat{S}_x^4|E_k\rangle. \tag{E.5}$$

The parameter regime $h \leqslant h_c^k$ allows us to exploit the degeneracy $E_k = \tilde{E}_k$, that implies $\partial_h\big(E_k^2/h\big) = \partial_h\big(\tilde{E}_k^2/h\big)$, and the structure of Hamiltonian eigenstates in the $S_x$ eigenbasis [52], that entails $\langle E_k|\hat{S}_x^4|E_k\rangle \simeq \langle \tilde{E}_k|\hat{S}_x^4|\tilde{E}_k\rangle$. From these properties and from equation (E.5), we obtain $\langle E_k|\hat{S}_z^2|E_k\rangle \simeq \langle \tilde{E}_k|\hat{S}_z^2|\tilde{E}_k\rangle$. Recalling equation (E.2), we also conclude that $\Delta^2_{|E_k\rangle}S_z$ and $\Delta^2_{|\tilde{E}_k\rangle}S_z$ have the same scaling with $N$.

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
