# Peer review of "Precision magnetometry exploiting excited state quantum phase transitions"

_SciPost Physics, doi:SciPost Phys. 17, 043 (2024)_

## Round 2 · Referee Report · Anonymous (Referee 1) · 2023-9-24

Report

Report - Precision magnetometry exploiting excited state quantum phase transitions
The authors report two novel metrological protocols for the estimation of the magnetic field with a particular critical spin system with an infinite-range interaction. Their approach is based on the exploitation of excited state quantum phase transitions (ESQPT), and considering they focus on finite-size models the criticality enhancing the metrological protocols relies on the crossings of excited levels.
The authors characterize the spectral density of the ESQPTs and found that are around the critical point of the model in the thermodynamical limit. Moreover, within this interval, each excited state has a unique critical field, and at these points, the Quantum Fisher Information (QFI), which measures measurement precision, exhibits broad peaks.These spectral properties of ESQPT are essential for two efficient magnetometry schemes. The first scheme achieves sub-shot-noise accuracy with a small quantum register and a constant running time. The second scheme offers high accuracy with a logarithmic number of register qubits and polynomial running time, outperforming conventional estimations with fewer qubits. Importantly, these ESQPT-based magnetometry protocols are robust against various noise sources due to the characteristics of QFI peaks. While entanglement is usually crucial in quantum metrology, here, the critical behavior of the states plays a pivotal role in achieving enhanced precision magnetometry.
The paper is well written and presents very interesting and new results, so I recommend the publication. Anyway, I have a few suggestions to improve the clarity of the manuscript.
It is not clear if the different colors have a specific meaning in Figure 1a. If yes, can the authors please explain it?
Can please the author comment which techniques they use to compute the spectrum, even in the sector s = N/2 with dimension N+1 for the considered systems with a big number of spins, e.g., N=12000?
Reorganize the citations, avoid multiple citations to facilitate the readability. For example at the end of the second paragraph of page 4, the citation [50, 27, 51, 52, 23, 53, 47, 54, 27] can be rearranged to have [23, 27, 47, 50-54]
Is it possible to give the analytical expression of the vertical critical lines in the caption of Figure 2?
I think there is an extra (d) In the second line in the caption of Figure 2, and that the expression ``critical fields for two adjacent excited states’’ should be explicated.
I would suggest to rewrite subsection 4.1.1 about probe preparation, it is less clear than the rest of the manuscript.
I would also like to point out that (ground state and excited states) phase transitions of such type of critical systems are nowadays feasibly simulated on NISQ-hardware, see e.g., Phys. Rev. E 107, 024113 (2023).
  • validity: -
  • significance: -
  • originality: -
  • clarity: -
  • formatting: -
  • grammar: -

Author:  Ugo Marzolino  on 2023-10-12  [id 4036]

(in reply to Report 1 on 2023-09-24)

We are very glad that the reviewer considers our manuscript well-written and interesting. We have addressed the issues raised by the reviewer, and listed the changes in the following.

Reviewer:
It is not clear if the different colors have a specific meaning in Figure 1a. If yes, can the authors please explain it?

Answer:
The colours in figure 1(a) do not have a specific meaning, but are intended to make the figure clearer for small magnetic fields when the curves become closer.

Reviewer:
Can please the author comment which techniques they use to compute the spectrum, even in the sector $s=N/2$ with dimension $N+1$ for the considered systems with a big number of spins, e.g., $N=12000$?

Answer:
We have computed the spectrum of the Hamiltonian in the sectors with $s=N/2$ and fixed parity (either even or odd) using exact diagonalization using MatLab 2018b and the command “eig”. The time required for exact diagonalization is of the order of several minutes even for N=12000. We have added the following sentence five lines after equation (1): “These symmetries allow us to compute numerical exact diagonaliziation of the LMG Hamiltonian in the orthogonal subspaces with $s=N/2$ and even or odd parity”.

Reviewer:
Reorganize the citations, avoid multiple citations to facilitate the readability. For example at the end of the second paragraph of page 4, the citation [50, 27, 51, 52, 23, 53, 47, 54, 27] can be rearranged to have [23, 27, 47, 50-54].

Answer:
The document class and the bibtex style in the previous version were responsible of the order of citations. After using the SciPost template, the citations are organized as the reviewer suggested.

Reviewer:
Is it possible to give the analytical expression of the vertical critical lines in the caption of Figure 2?

Answer:
We have added the analytical expression of vertical the lines in the caption of figure 2. These lines correspond to the critical fields $h_c^k$ of different eigenstates ($|E_k\rangle$ and $|\tilde E_k\rangle$), namely the field value such that $E_k=E_c=-Nh_c^k/2$. Therefore, $h_c^k=-2E_k/N$. The analytical expression of the eigenvalues $E_k$ is not known in general.

Reviewer:
I think there is an extra (d) In the second line in the caption of Figure 2, and that the expression critical fields for two adjacent excited states’’ should be explicated.

Answer:
We thank the reviewer for noticing this typo. Indeed, the first of the (d) has been replaced with (b). The expression “critical fields for two adjacent excited states” appeared in the paragraph starting with “We further compare…” at page 6. We have re-written that sentence. We have also replaced the term “adjacent eigenenergy gap $\Delta E=E_{k+1}-E_{k+1}$” with “eigenenergy gap $\Delta E=E_{k+1}-E_{k+1}$” in the caption of figure 2.

Reviewer:
I would suggest to rewrite subsection 4.1.1 about probe preparation, it is less clear than the rest of the manuscript.

Answer:
We have re-written subsection 4.1.1. In particular, we slightly changed the first paragraph in order to be a little bit more explicit concerning the reduction to the subsystem with $s=N/2$, and we have added several details on the phase estimation algorithm used for probe preparation in the subsequent paragraphs.

Reviewer:
I would also like to point out that (ground state and excited states) phase transitions of such type of critical systems are nowadays feasibly simulated on NISQ-hardware, see e.g., Phys. Rev. E 107, 024113 (2023).

Answer:
We thank the referee for bringing the interesting paper Phys. Rev. E 107, 024113 (2023) to our attention. It is indeed a promising study of simultations of the LMG eigenstates with measurements of the corresponding phase transitions on NISQ device. We have introduced very shortly the context and the terminology of NISQ technologies sentence in the introduction. We have then mentioned in the conclusions that the paper Phys. Rev. E 107, 024113 (2023), together with experimental platforms for realizing the LMG model, may inspire new implementations with NISQ devices.

---

## Round 3 · Referee Report · Anonymous (Referee 1) · 2023-10-20

Report

The authors have replied in a satisfactory manner to the points raised by the referees. The paper is now suitable for publication.

---

## Round 3 · Referee Report · Anonymous (Referee 2) · 2024-1-16

Strengths

1) Detailed presentation of the excited-state structure of the LMG, and analysis thereof through quantum metrological concepts. 2) Presentation of new protocols allowing to prepare states showing superextensive sensitivity

Weaknesses

Unclear whether the superextensivity really comes from the presence of the ESQPT: see attached report.

Report

I believe this work should be suitable for publication in Scipost, but I believe some of its claims may have to be amended; in particular, I am not sure whether the super-extensive scaling reported here is really a signature of an ESQPT, or simply a generic properties of highly-excited spin models (see attached report for details).

Once these points have been adressed, I will be happy to recommend this work for publication

Requested changes

See attached report.

Attachment

  • validity: high
  • significance: good
  • originality: high
  • clarity: high
  • formatting: -
  • grammar: -

Author:  Ugo Marzolino  on 2024-03-05  [id 4337]

(in reply to Report 2 on 2024-01-16)

We would like to thank very much the reviewer first of all for her/his accurate and stimulating comments, and then also for the overall positive assessment of our results and of their importance. We have resubmitted the manuscript amended according to all the reviewer’s suggestions.

Attachment:

reply_to_the_second_report_otUTb3p.pdf

---

## Round 3 · Author Response

We are very glad that the reviewer considers our manuscript well-written and interesting. We have addressed the issues raised by the reviewer, and listed the changes in the following.

Reviewer:
It is not clear if the different colors have a specific meaning in Figure 1a. If yes, can the authors please explain it?

Answer:
The colours in figure 1(a) do not have a specific meaning, but are intended to make the figure clearer for small magnetic fields when the curves become closer.

Reviewer:
Can please the author comment which techniques they use to compute the spectrum, even in the sector $s=N/2$ with dimension $N+1$ for the considered systems with a big number of spins, e.g., $N=12000$?

Answer:
We have computed the spectrum of the Hamiltonian in the sectors with $s=N/2$ and fixed parity (either even or odd) using exact diagonalization using MatLab 2018b and the command “eig”. The time required for exact diagonalization is of the order of several minutes even for N=12000. We have added the following sentence five lines after equation (1): “These symmetries allow us to compute numerical exact diagonaliziation of the LMG Hamiltonian in the orthogonal subspaces with $s=N/2$ and even or odd parity”.

Reviewer:
Reorganize the citations, avoid multiple citations to facilitate the readability. For example at the end of the second paragraph of page 4, the citation [50, 27, 51, 52, 23, 53, 47, 54, 27] can be rearranged to have [23, 27, 47, 50-54].

Answer:
The document class and the bibtex style in the previous version were responsible of the order of citations. After using the SciPost template, the citations are organized as the reviewer suggested.

Reviewer:
Is it possible to give the analytical expression of the vertical critical lines in the caption of Figure 2?

Answer:
We have added the analytical expression of vertical the lines in the caption of figure 2. These lines correspond to the critical fields $h_c^k$ of different eigenstates ($|E_k\rangle$ and $|\tilde E_k\rangle$), namely the field value such that $E_k=E_c=-Nh_c^k/2$. Therefore, $h_c^k=-2E_k/N$. The analytical expression of the eigenvalues $E_k$ is not known in general.

Reviewer:
I think there is an extra (d) In the second line in the caption of Figure 2, and that the expression critical fields for two adjacent excited states’’ should be explicated.

Answer:
We thank the reviewer for noticing this typo. Indeed, the first of the (d) has been replaced with (b). The expression “critical fields for two adjacent excited states” appeared in the paragraph starting with “We further compare…” at page 6. We have re-written that sentence. We have also replaced the term “adjacent eigenenergy gap $\Delta E=E_{k+1}-E_{k+1}$” with “eigenenergy gap $\Delta E=E_{k+1}-E_{k+1}$” in the caption of figure 2.

Reviewer:
I would suggest to rewrite subsection 4.1.1 about probe preparation, it is less clear than the rest of the manuscript.

Answer:
We have re-written subsection 4.1.1. In particular, we slightly changed the first paragraph in order to be a little bit more explicit concerning the reduction to the subsystem with $s=N/2$, and we have added several details on the phase estimation algorithm used for probe preparation in the subsequent paragraphs.

Reviewer:
I would also like to point out that (ground state and excited states) phase transitions of such type of critical systems are nowadays feasibly simulated on NISQ-hardware, see e.g., Phys. Rev. E 107, 024113 (2023).

Answer:
We thank the referee for bringing the interesting paper Phys. Rev. E 107, 024113 (2023) to our attention. It is indeed a promising study of simultations of the LMG eigenstates with measurements of the corresponding phase transitions on NISQ device. We have introduced very shortly the context and the terminology of NISQ technologies sentence in the introduction. We have then mentioned in the conclusions that the paper Phys. Rev. E 107, 024113 (2023), together with experimental platforms for realizing the LMG model, may inspire new implementations with NISQ devices.

---

## Round 3 · List of Changes

• We have added the following sentence five lines after equation (1): “These symmetries allow us to compute numerical exact diagonaliziation of the LMG Hamiltonian in the orthogonal subspaces with s=N/2 and even or odd parity”.

  • We have used the SciPost template, and the citations are organized as the reviewer suggested.

  • We have added the analytical expression of vertical the lines in the caption of figure 2.

  • We have re-written the sentence starting with “We further compare…” at page 6.

  • We have re-written subsection 4.1.1. In particular, we slightly changed the first paragraph in order to be a little bit more explicit concerning the reduction to the subsystem with s=N/2, and we have added several details on the phase estimation algorithm used for probe preparation in the subsequent paragraphs.

  • We have added the following sentence in the introduction “It is also desirable to investigate metrological schemes suited for several physical platforms and using different physical phenomena in the search for feasible implementations, so called noisy intermediate-scale quantum (NISQ) technologies [15,16]” in order to introduce the context and the terminology. We have also and added the following sentence in the conclusions “Eingenstates of the LMG model at small size can be simulated also with variational hybrid quantum-classical algorithms with low depth on superconducting transmon qubits, and signatures of their phase transitions can be computed [112]” after mentioning other realizations of the LMG model, and conluded with “The robustness of magnetometric protocols based on the ESQPT and the aforementioned variety of platforms for realizing or simulating the LMG model make these protocols good candidates for NISQ devices”.

---

## Round 4 · Referee Report · Anonymous (Referee 2) · 2024-3-13

Report

The authors have made some changes to the manuscript, amending some of the claims which I considered problematic, and improved the overall clarity of the paper. There are, however, some points which I still considered unresolved, and that need adressed before I can recommand this paper for publication (see attached report)

Requested changes

Plot F/N^2 instead of F in Fig.4, in order to determine whether superextensivity can be related to the ESQPT. Amend some of the claims (see Report).

Attachment

  • validity: -
  • significance: -
  • originality: -
  • clarity: -
  • formatting: -
  • grammar: -

Author:  Ugo Marzolino  on 2024-07-08  [id 4606]

(in reply to Report 1 on 2024-03-13)
Category:
answer to question

Dear editor and reviewer,

we understand the referee's issue, and therefore we have amended the claims according to the referee's suggestions. We also drawn new plots as suggested by the referee concerning the scaling of the quantum Fisher information. We do not enter the question whether these scalings are signature of the excited state quantum phase transition, as this is not the scope of our manuscript. Moreover, we do not think that this discussion is fundamental for presenting out results in the field of quantum metrology, as already acknowledged by the referee. For these reasons and in order to not weight down the main discussion, we commented on the new figures in appendix B and we amended the claims as suggested by the referee. Further details are described in the attached file.

We think that we have addressed the criticisms of the referee, and that the new resubmitted manuscript is suitable for publication.

Yours sincerely,

Qian Wang and Ugo Marzolino

Attachment:

answer_to_the_third_referee_report.pdf

---

## Round 4 · Author Response

We would like to thank very much the reviewer first of all for her/his accurate and stimulating comments, and then also for the overall positive assessment of our results and of their importance. We have resubmitted the manuscript amended according to all the reviewer’s suggestions as described in the file attached to our response to the report.

---

## Round 5 · Referee Report · Anonymous (Referee 5) · 2024-7-15

Report

The authors have modified the manuscript following my recommendations. In particular, the new Fig.11 seems to confirm the point, I made, ie, that the super-extensive scaling observed here is a property of the QFI for higher-excited state, but is not directly related to the presence of an ESQPT.

I still believe that the origin of the super-extensive scaling observed here is an important issue which has not been fully adressed in the manuscript, and I think it would have been more transparent to put the new Fig.11 in the main text, instead of relegated in an Appendix. However, given the changes that have already been made, and in order not to drag things down too long, I recommand publication of the manuscript as it stands.

Recommendation

Publish (meets expectations and criteria for this Journal)

---

## Round 5 · Author Response

Dear editor and reviewer,

we understand the referee's issue, and therefore we have amended the claims according to the referee's suggestions. We also drawn new plots as suggested by the referee. The plot of $F/N^2$ requested in the report is the new figure 11. In the report attachment, the referee also asks for the plot of $\mathcal{F}_{h,m}/N^2$ instead of $\mathcal{F}_{h,m}$ in figure 4. We guess the referee meant $\mathcal{F}_{h,m}^*$ as this is the quantity plotted in figure 4. $\mathcal{F}_{h,m}^*$ in figure 4 (as well as $\mathcal{F}_{h,m}$ in figure 3) is plotted as a function of $N$ in log-log scale: there are not curves $\mathcal{F}_{h,m}^*$ and $\mathcal{F}_{h,m}$ for different $N$. The plot for $\mathcal{F}_{h,m}^*/N^2$ similar to that in figure 4 is simply the same as $\mathcal{F}_{h,m}^*$ with slope $\xi-2$ instead of $\xi$. We therefore plotted $\mathcal{F}_{h,m}^*/N^2$ as function of $h$ and $\mathcal{F}_{h,m}/N^2$ as function of the eigenenergy for different $N$ in figure 12. Notice that $\mathcal{F}_{h,m}$ is the maximum of $\mathcal{F}_h$ over all $h$ (the peak value exemplified in figures 2(a) that corresponds to fixing $h=h_c^k$), then it depends only on the eigenenergy and on $N$. Similarly, $\mathcal{F}_{h,m}^*$ is the maximum of $\mathcal{F}_h$ over all eigenenergies (the peak value exemplified in figures 4(a,b) with $E_k=E_c$), then it depends only on $h$ and on $N$.

Figures 11 and 12 show that $\mathcal{F}_h/N^2$ overlap for different $N$ except around the peak, whose value slowly increases with $N$. This is compatible with the fit of the peak values $\mathcal{F}_{h,m}$ and $\mathcal{F}_{h,m}^*$ which show a power law with exponent slightly larger than $2$. We do not enter the question whether these scalings are signature of the excited state quantum phase transition, as this is not the scope of our manuscript. Moreover, we do not think that this discussion is fundamental for presenting out results in the field of quantum metrology, as already acknowledged by the referee. For these reasons and in order to not weight down the main discussion, we commented on the new figures in appendix B and we amended the claims as suggested by the referee.

We think that we have addressed the criticisms of the referee, and that the new resubmitted manuscript is suitable for publication.

Yours sincerely,

Qian Wang and Ugo Marzolino

---

## Round 5 · List of Changes

\begin{itemize}
\item The sentence "$\mathcal{F}_h$ exhibits a sharp peak close to the critical energy $E_c$, and its maximum value [...] increases with the system size $N$" has been replace by "$\mathcal{F}_h$ exhibits a sharp peak close to the critical energy $E_c$ [...]", dropping the reference to the size scaling, and we have consistely reworded the beginning of the next sentence.

\item The sentence "We suggest that the superextensive peaks of the QFI [...] are a signature of the ESQPT" has been removed, and we have consistely reworded the beginning of the paragraph.

\item We have changed the title of appendix B, and discussed the new figures 11 and 12 there.
\end{itemize}

---

## Editorial Decision

published